# On Random Deep Weight-Tied Autoencoders: Exact Asymptotic Analysis, Phase Transitions, and Implications to Training

**Ping Li,**[*] **Phan-Minh Nguyen**[†]

## Abstract

We study the behavior of weight-tied multilayer vanilla autoencoders under the assumption of random weights. Via an exact characterization in the limit of large dimensions, our analysis reveals interesting phase transition phenomena when the depth becomes large. This, in particular, provides quantitative answers and insights to three questions that were yet fully understood in the literature. Firstly, we provide a precise answer on how the random deep weight-tied autoencoder model performs "approximate inference" as posed by Scellier et al. (2018), and its connection to reversibility considered by several theoretical studies. Secondly, we show that deep autoencoders display a higher degree of sensitivity to perturbations in the parameters, distinct from the shallow counterparts. Thirdly, we obtain insights on pitfalls in training initialization practice, and demonstrate experimentally that it is possible to train a deep autoencoder, even with the tanh activation and a depth as large as 200 layers, without resorting to techniques such as layer-wise pre-training or batch normalization. Our analysis is not specific to any depths or any Lipschitz activations, and our analytical techniques may have broader applicability.

## 1 Introduction

The autoencoder is a cornerstone in machine learning, first as a response to the unsupervised learning problem (Rumelhart & Zipser (1985)), then with applications to dimensionality reduction (Hinton & Salakhutdinov (2006)), unsupervised pre-training (Erhan et al. (2010)), and also as a precursor to many modern generative models (Goodfellow et al. (2016)). Its reconstruction power is well utilized in applications such as anomaly detection (Chandola et al. (2009)) and image recovery (Mousavi et al. (2015)). With the surge of deep learning, thousands of papers have studied multilayer variants of this architecture, but theoretical understanding has been limited, since analyzing the learning dynamics of a highly nonlinear structure is typically a difficult problem even for the shallow autoencoder. To get around this, we tackle the task with a critical assumption: the weights are random and the autoencoder is weight-tied. One enjoys much analytical tractability from the randomness assumption, whereas weight tying enforces the random autoencoder to perform "autoencoding". We also study this in the high-dimensional setting, where all dimensions are comparably large and ideally jointly approaching infinity. We consider the simplest setting: vanilla autoencoders (i.e., ones with fully connected layers only) and their reconstruction capability. This is done for the sake of understanding the effect of depth, while we note our techniques may have broader applicability.

The aforementioned assumptions are not without justifications. There is a growing literature on deep neural networks with random weights, (Li & Saad (2018); Giryes et al. (2016); Poole et al. (2016); Schoenholz et al. (2016); Gabrié et al. (2018); Amari et al. (2018)) to name a few, revealing certain properties of deep feedforward networks[1]. Several recent works have also studied random multilayer feedforward networks through the lens of statistical inference (Manoel et al. (2017);

---

[*]Cognitive Computing Lab (CCL), Baidu Research, Bellevue, WA 98004, USA. liping11@baidu.com.

[†]Stanford University, Stanford, CA 94305, USA. pm.nguyen@stanford.edu. Work performed at Baidu Research as Summer Intern in 2018.

[1]It is worth mentioning that several characteristics of deep random feedforward networks were implicitly investigated a few decades ago, in the form of (continuous- or discrete-time) recurrent dynamics of shallow random neural networks (Sompolinsky et al. (1988); Cessac et al. (1994); Cessac (1995)) under the "local chaos" hypothesis (Amari (1972)).

Reeves (2017); Fletcher et al. (2018)). The idea of weight tying is considered in the important paper Vincent et al. (2010) with an empirical finding that autoencoders with and without weight tying perform comparably, and has become standard in autoencoders. Similar features of random connection and symmetry also appear in other neural models (Lillicrap et al. (2016); Scellier et al. (2018)). Finally the high-dimensional setting is common in recent statistical learning advances (Bühlmann & Van De Geer (2011)), and not too far from the actual practice where many large datasets have dimensions of at least a few hundreds and are harnessed by large-scaled models.

We seek quantitative answers to three specific questions that are motivated by previous works:

- In exactly what way does the (vanilla) random weight-tied autoencoder perform "approximate inference"? This term is coined in Scellier et al. (2018) in connection with the theoretical results in Arora et al. (2015), which implicitly studies the said model. In particular, Arora et al. (2015) proves an upper bound on $\|\hat{x} - x\|^2$, where $x$ and $\hat{x}$ are the input and the output of the network, but is limited in the number of layers and specific to the ReLU activation. This direction has been recently extended by Gilbert et al. (2017). In our work, we establish precisely what this approximate inference is by obtaining a general and asymptotically exact characterization[2] of $\hat{x}$, for any number of layers and any Lipschitz continuous activations (Theorem 1 and Section 3.3). Theorem 1 is the key theoretical result of our work and lays the foundation for all analyses that follow.

- In what way is the deep autoencoder different from the shallow counterpart? Li & Saad (2018); Poole et al. (2016) reveal this in terms of the candidate function space and expressivity for feedforward networks. It is unclear how these notions are applicable to weight-tied autoencoders, which seek replication of the input rather than a generic mapping. In this work, we show that the deep autoencoder exhibits a higher order of sensitivity to perturbations of the parameters (Section 3.4).

- Does it have any implications to the training practice? Many recent works[3] Glorot & Bengio (2010); He et al. (2015); Schoenholz et al. (2016); Pennington et al. (2017); Yang & Schoenholz (2017); Xiao et al. (2018); Chen et al. (2018); Hayou et al. (2018); Hanin & Rolnick (2018); Hanin (2018); Burkholz & Dubatovka (2018) demonstrate a connection between the study of random networks, or ones at initialization, and their trainability. Note that these works either do not study weight-tied structures, or assume the analysis of the untying case for weight-tied structures. In our work, we derive and experimentally verify insights on how (not) to initialize deep weight-tied autoencoders, demonstrating that it is possible to train them without resorting to techniques such as greedy layer-wise pretraining, drop-out and batch normalization (Section 3.5). Specifically we experiment with 200-layer autoencoders.

No prior works have attempted all three tasks. The quantitative difference between weight-tied and weight-untied networks is in fact not negligible, yet the analysis is non-trivial due to the weight tying constraint (Arora et al. (2015); Chen et al. (2018)). To address this issue and obtain Theorem 1, we apply the Gaussian conditioning technique, which first appears in the studies of TAP equations in spin glass theory (Bolthausen (2014)) and is extensively used in the approximate message passing algorithm literature (Bayati & Montanari (2011); Javanmard & Montanari (2013); Berthier et al. (2017)). This should be contrasted with untied random networks, whose analysis is typically more straightforward. More importantly, the difference is not only analytical: the overall picture of deep random weight-tied autoencoders is rich and drastically different from that of feedforward networks. An analysis in the limit of infinite depth reveals three fundamental equations governing the picture (Section 3.1), which displays multiple phase transition phenomena (Section 3.2).

Finally let us quickly mention other recent theoretical works on autoencoders: Arora et al. (2014); Arpit et al. (2015); Rangamani et al. (2017); Nguyen et al. (2018) studying the learned autoencoders in specific settings, Baldi (2012); Alain & Bengio (2014); Bengio et al. (2013) taking a framework where the encoder and the decoder are generic mappings, and Le Roux & Bengio (2008); Sutskever

---

[2]Roughly speaking, we prove that $\hat{x} \cong c_1 x + c_2 z$, for $c_1$ and $c_2$ scalars and $z$ an independent Gaussian vector. It is then straightforward to deduce $\|\hat{x} - x\|^2$. This quantity is hence meaningful when $c_1 = 1$.

[3]Many of these works refer to the random network framework as the mean-field analysis. Here we avoid using the term "mean-field" since it is also used to refer to the analysis of the learned infinitely-wide neural networks (Mei et al. (2018)), which is a different setting.

& Hinton (2008); Montufar & Ay (2011) exploring representational properties of related architectures. These works head in different directions from ours.

## 2    SETTING AND MAIN THEOREM

### 2.1    SETTING AND ASSUMPTIONS

Consider the following $2L$-layers autoencoder with weight tying:

$$\hat{\boldsymbol{x}} = \varphi_0 \left( \boldsymbol{W}_1^\top \varphi_1 \left( ... \varphi_{L-1} \left( \boldsymbol{W}_L^\top \varphi_L \left( \boldsymbol{W}_L \sigma_{L-1} \left( ... \sigma_1 \left( \boldsymbol{W}_1 \sigma_0 \left( \boldsymbol{x} \right) + \boldsymbol{b}_1 \right) + ... \right) + \boldsymbol{b}_L \right) + \boldsymbol{v}_L \right) + ... \right) + \boldsymbol{v}_1 \right).$$

Here $\boldsymbol{x} \in \mathbb{R}^{n_0}$ is the input, $\boldsymbol{W}_\ell \in \mathbb{R}^{n_\ell \times n_{\ell-1}}$ is the weight, $\boldsymbol{b}_\ell \in \mathbb{R}^{n_\ell}$ is the encoder bias, and $\boldsymbol{v}_\ell \in \mathbb{R}^{n_{\ell-1}}$ is the decoder bias, for $\ell = 1, ..., L$. Also $\varphi_\ell : \mathbb{R} \mapsto \mathbb{R}$ and $\sigma_\ell : \mathbb{R} \mapsto \mathbb{R}$ are the activations (where for a vector $\boldsymbol{u} \in \mathbb{R}^n$ and a function $\varphi : \mathbb{R} \mapsto \mathbb{R}$, we write $\varphi(\boldsymbol{u})$ to denote the vector $\left( \varphi(u_1), ..., \varphi(u_n) \right)^\top$). It is usually the case in practice that $\sigma_0(\boldsymbol{u}) = \boldsymbol{u}$ the identity function. We introduce some convenient quantities inductively:

$$\boldsymbol{x}_0 = \boldsymbol{x}, \qquad \boldsymbol{x}_\ell = \boldsymbol{W}_\ell \sigma_{\ell-1}(\boldsymbol{x}_{\ell-1}) + \boldsymbol{b}_\ell, \quad \ell = 1, ..., L,$$

$$\hat{\boldsymbol{x}}_L = \boldsymbol{W}_L^\top \varphi_L(\boldsymbol{x}_L) + \boldsymbol{v}_L, \qquad \hat{\boldsymbol{x}}_\ell = \boldsymbol{W}_\ell^\top \varphi_\ell(\hat{\boldsymbol{x}}_{\ell+1}) + \boldsymbol{v}_\ell, \quad \ell = L-1, ..., 1.$$

Note that $\hat{\boldsymbol{x}} = \varphi_0(\hat{\boldsymbol{x}}_1)$. See Fig. 5 of Appendix A.1 for a schematic diagram. We assume weights are random. Specifically we generate the weights and biases according to

$$(\boldsymbol{W}_\ell)_{ij} \sim \mathcal{N}\left( 0, \frac{\sigma_{W,\ell}^2}{n_{\ell-1}} \right) \text{ i.i.d.,} \quad (\boldsymbol{b}_\ell)_i \sim \mathcal{N}\left( 0, \sigma_{b,\ell}^2 \right) \text{ i.i.d.,} \quad (\boldsymbol{v}_\ell)_i \sim \mathcal{N}\left( 0, \sigma_{v,\ell}^2 \right) \text{ i.i.d.}$$

independently of each other. The scaling of the variances accords with the literature and actual practice (Glorot & Bengio (2010); Vincent et al. (2010)). We also consider the asymptotic high-dimensional regime, indexed by $n$:

$$n_\ell = n_\ell(n), \quad \frac{n_\ell}{n_{\ell-1}} \to \alpha_\ell > 0 \text{ and } n_\ell \to \infty \text{ as } n \to \infty, \quad \forall \ell.$$

Here $\sigma_{W,\ell}$, $\sigma_{b,\ell}$, $\sigma_{v,\ell}$ and $\alpha_\ell$ are finite constants independent of $n$. We enforce $\sigma_{W,\ell} > 0$, but allow $\sigma_{b,\ell}$ and $\sigma_{v,\ell}$ to be zero. We assume that all activations are Lipschitz continuous, and the encoder activations $\sigma_\ell$'s are non-trivial in the sense that for any $\tau > 0$, $\mathbb{E}_z\left\{ \sigma_\ell(\tau z)^2 \right\} > 0$ where $z \sim \mathcal{N}(0, 1)$. We also assume that $\frac{1}{n_0}\|\sigma_0(\boldsymbol{x})\|^2$ tends to a finite and strictly positive constant as $n \to \infty$. We refer to Appendix A.1 for more clarifications of notations.

### 2.2    MAIN THEOREM

We motivate our main result via a simplified shallow autoencoder: $\hat{\boldsymbol{x}} = \boldsymbol{W}^\top \varphi(\boldsymbol{W}\boldsymbol{x}) = \sum_{i=1}^m \boldsymbol{w}_i \varphi(\boldsymbol{w}_i^\top \boldsymbol{x})$, where $\boldsymbol{W} \in \mathbb{R}^{m \times n}$, $\boldsymbol{w}_i \in \mathbb{R}^n$. Notice $\hat{\boldsymbol{x}}$ is a sum of independent terms, and by Stein's lemma (cf. Appendix E.2), $\mathbb{E}\{\hat{\boldsymbol{x}}\} \propto \mathbb{E}\left\{ \sum_{i=1}^m \varphi'(\boldsymbol{w}_i^\top \boldsymbol{x})/m \right\} \boldsymbol{x}$. One thus expects $\hat{\boldsymbol{x}} \approx c_1 \boldsymbol{x} + c_2 \boldsymbol{z}$ for scalars $c_1$, $c_2$ and $\boldsymbol{z} \sim \mathcal{N}(0, \boldsymbol{I}_n)$, for large $n$ and $m$. It is then important to specify exactly $c_1$ and $c_2$. Theorem 1 formalizes this intuition with precise formulas for the scalars.

We now define some scalar sequences, which will then be related to the (vector) quantities of the autoencoders in Theorem 1. First we define $\{\tau_\ell\}_{\ell=1,...,L}$ and $\{\bar{\tau}_\ell\}_{\ell=0,...,L}$ inductively:

$$\bar{\tau}_0^2 = \frac{1}{n_0}\|\sigma_0(x)\|^2, \qquad \bar{\tau}_\ell^2 = \tau_\ell^2 + \sigma_{b,\ell}^2, \quad \ell = 1, ..., L,$$

$$\tau_1^2 = \sigma_{W,1}^2 \bar{\tau}_0^2, \qquad \tau_\ell^2 = \sigma_{W,\ell}^2 \mathbb{E}_z\left\{ \sigma_{\ell-1}(\bar{\tau}_{\ell-1}z)^2 \right\}, \quad \ell = 2, ..., L,$$

for $z \sim \mathcal{N}(0, 1)$. Next, we define $\{\gamma_\ell, \rho_\ell\}_{\ell=2,...,L+1}$ inductively:

$$\gamma_{L+1} = \frac{1}{\bar{\tau}_L^2}\mathbb{E}_{z_1}\left\{ \bar{\tau}_L z_1 \varphi_L(\bar{\tau}_L z_1) \right\}, \qquad \rho_{L+1} = \mathbb{E}_{z_1}\left\{ \varphi_L(\bar{\tau}_L z_1)^2 \right\},$$

$$\gamma_\ell = \frac{1}{\bar{\tau}_{\ell-1}^2}\mathbb{E}_{z_1, z_2}\left\{ \bar{\tau}_{\ell-1}z_1 \varphi_{\ell-1}\left( \alpha_\ell \sigma_{W,\ell}^2 \gamma_{\ell+1}\sigma_{\ell-1}(\bar{\tau}_{\ell-1}z_1) + \sqrt{\alpha_\ell \sigma_{W,\ell}^2 \rho_{\ell+1} + \sigma_{v,\ell}^2}z_2 \right) \right\},$$

$$\rho_\ell = \mathbb{E}_{z_1, z_2}\left\{ \varphi_{\ell-1}\left( \alpha_\ell \sigma_{W,\ell}^2 \gamma_{\ell+1}\sigma_{\ell-1}(\bar{\tau}_{\ell-1}z_1) + \sqrt{\alpha_\ell \sigma_{W,\ell}^2 \rho_{\ell+1} + \sigma_{v,\ell}^2}z_2 \right)^2 \right\}, \quad \ell = L-1, ..., 2,$$

for $z_1, z_2 \sim \mathcal{N}(0, 1)$ independently. With these sequences defined, we can state the main theorem. Its statement uses the relational operator $\cong$, which is defined formally in Appendix A.1. Roughly speaking, $\boldsymbol{a}_n \cong \boldsymbol{b}_n$ means $\boldsymbol{a}_n$ and $\boldsymbol{b}_n$ are asymptotically equal in distribution as $n \to \infty$.

**Theorem 1.** *Consider the settings and assumptions as in Section 2.1, and the sequences $\{\tau_\ell, \bar{\tau}_\ell\}$ and $\{\gamma_\ell, \rho_\ell\}$ defined as above. Then in the limit $n \to \infty$:*

(a) *$\{\bar{\tau}_\ell\}$ describes the behavior of the encoder output $\boldsymbol{x}_\ell$:*

$$\boldsymbol{x}_\ell \cong \bar{\tau}_\ell \boldsymbol{z}, \quad \ell = 1, ..., L,$$

*for $\boldsymbol{z} \sim \mathcal{N}(0, \boldsymbol{I}_{n_\ell})$.*

(b) *$\{\bar{\tau}_\ell, \gamma_\ell, \rho_\ell\}$ describes the behavior of the decoder output $\hat{\boldsymbol{x}}_\ell$:*

$$\hat{\boldsymbol{x}}_\ell \cong \alpha_\ell \sigma_{W,\ell}^2 \gamma_{\ell+1} \sigma_{\ell-1} (\bar{\tau}_{\ell-1} \boldsymbol{z}_1) + \sqrt{\alpha_\ell \sigma_{W,\ell}^2 \rho_{\ell+1} + \sigma_{v,\ell}^2} \boldsymbol{z}_2, \quad \ell = 2, ..., L,$$

*for $\boldsymbol{z}_1, \boldsymbol{z}_2 \sim \mathcal{N}(0, \boldsymbol{I}_{n_{\ell-1}})$ independently. One can replace $\bar{\tau}_{\ell-1} \boldsymbol{z}_1$ with $\boldsymbol{x}_{\ell-1}$ in the above, with $\boldsymbol{z}_2$ independent of $\boldsymbol{x}_{\ell-1}$, in which case the statement also holds for $\ell = 1$ with $\boldsymbol{x}_0 = \boldsymbol{x}$.*

(c) *For the autoencoder's output $\hat{\boldsymbol{x}}$,*

$$\hat{\boldsymbol{x}} \cong \varphi_0 \left( \alpha_1 \sigma_{W,1}^2 \gamma_2 \sigma_0 (\boldsymbol{x}) + \sqrt{\alpha_1 \sigma_{W,1}^2 \rho_2 + \sigma_{v,1}^2} \boldsymbol{z}_2 \right),$$

*for $\boldsymbol{z}_2 \sim \mathcal{N}(0, \boldsymbol{I}_{n_0})$ independent of $\boldsymbol{x}$.*

The proof of the theorem, as well as an outline of the key ideas, are in Appendix A. The theorem says that $\boldsymbol{x}_\ell$, $\hat{\boldsymbol{x}}_\ell$ and $\hat{\boldsymbol{x}}$ admit simple descriptions which are tracked by scalar sequences $\{\bar{\tau}_\ell, \gamma_\ell, \rho_\ell\}$. Hence we can learn about the autoencoder by analyzing $\{\bar{\tau}_\ell, \gamma_\ell, \rho_\ell\}$, which is generally a simpler task than studying $\boldsymbol{x}_\ell$, $\hat{\boldsymbol{x}}_\ell$ and $\hat{\boldsymbol{x}}$ directly. Numerical simulations in Appendix B suggest that, although the theorem's statement is in the infinite dimension limit, the agreement is already good for dimensions of a few hundreds. We note that while the theorem assumes Gaussian biases, the same proof technique allows to obtain a similar result with a more relaxed condition on the biases.

*Remark* 2. While the theorem is specific to $\boldsymbol{W}_\ell$ following the Gaussian distribution, simulations in Appendix B suggest that the conclusion holds for a much broader class of distributions. We conjecture that it should hold so long as each $\boldsymbol{W}_\ell$ has i.i.d. entries and is independent of each other, its distribution has bounded $k$-th moment for some sufficiently large $k$, and the activations as well as the input $\boldsymbol{x}$ satisfy certain mild regularity conditions.

*Remark* 3. We comment on the range of $\rho_\ell$ and $\gamma_\ell$. We have $\rho_\ell \geq 0$, which is obvious, and if $\|\varphi_{\ell-1}\|_\infty \leq C$, then $\rho_\ell \leq C^2$. By Stein's lemma (cf. Appendix E.2),

$$\gamma_{L+1} = \mathbb{E}_z \{\varphi_L'(\bar{\tau}_L z_1)\},$$

$$\gamma_\ell = \alpha_\ell \sigma_{W,\ell}^2 \gamma_{\ell+1} \mathbb{E}_{z_1, z_2} \left\{ \varphi_{\ell-1}' \left( \alpha_\ell \sigma_{W,\ell}^2 \gamma_{\ell+1} \sigma_{\ell-1} (\bar{\tau}_{\ell-1} z_1) + \sqrt{\alpha_\ell \sigma_{W,\ell}^2 \rho_{\ell+1} + \sigma_{v,\ell}^2} z_2 \right) \sigma_{\ell-1}' (\bar{\tau}_{\ell-1} z_1) \right\}.$$

If the activations are non-decreasing, then $\gamma_\ell \geq 0$. Furthermore, if the activations are Lipschitz, then $|\gamma_\ell| \leq C c^\ell$ for some constants $C$ and $c$.

## 3 AN ANALYSIS AT INFINITE DEPTH

In the following, we adopt a semi-rigorous approach, with an emphasis on the overall picture.

### 3.1 INFINITE DEPTH SIMPLIFICATION

We make several analytical simplifications. First consider $\alpha_\ell = \alpha > 0$, $\sigma_{W,\ell}^2 = \sigma_W^2 > 0$, $\sigma_{b,\ell}^2 = \sigma_b^2 \geq 0$, $\varphi_\ell = \varphi$ and $\sigma_\ell = \sigma$ all independent of $\ell$, except for $\varphi_L$ which is chosen separately (but we shall see that the specific choice of $\varphi_L$ is largely immaterial). We also assume that $\sigma_{v,\ell}^2 = 0$, and $\sigma_0$ and $\varphi_0$ are the identity[4]. We introduce a parameter $\bar{\tau} \geq 0$, whose role will be clear shortly, and which satisfies:

$$\bar{\tau}^2 = T(\bar{\tau}^2) \equiv T(\bar{\tau}^2; \sigma_W^2, \sigma_b^2, \sigma), \qquad T(\bar{\tau}^2) = \sigma_W^2 \mathbb{E}\left\{ \sigma(\bar{\tau} z)^2 \right\} + \sigma_b^2, \tag{1}$$

---

[4]The assumption $\sigma_{v,\ell}^2 = 0$ aligns well with several recent theoretical analyses of the autoencoders (Arpit et al. (2015); Rangamani et al. (2017); Gilbert et al. (2017); Nguyen et al. (2018)) which disregard the decoder's

for $z \sim \mathcal{N}(0,1)$. Note that this implies $\sigma_W^2 \leq \sigma_{W,\max}^2 = \bar{\tau}^2 / \mathbb{E}\left\{\sigma(\bar{\tau}z)^2\right\}$. If this cannot be satisfied, we set $\bar{\tau}^2 = +\infty$. In addition, let $\bar{\beta} = \alpha\sigma_W^2 > 0$. With these, let us consider the following two fixed point equations:

$$\gamma = G(\gamma, \rho) \equiv G\left(\gamma, \rho; \beta, \bar{\tau}^2, \sigma, \varphi\right), \qquad G(\gamma, \rho) = \frac{1}{\bar{\tau}^2}\mathbb{E}\left\{\bar{\tau}z_1\varphi\left(\beta\gamma\sigma(\bar{\tau}z_1) + \sqrt{\beta}\rho z_2\right)\right\}, \quad (2)$$

$$\rho = R(\gamma, \rho) \equiv R\left(\gamma, \rho; \beta, \bar{\tau}^2, \sigma, \varphi\right), \qquad R(\gamma, \rho) = \mathbb{E}\left\{\varphi\left(\beta\gamma\sigma(\bar{\tau}z_1) + \sqrt{\beta}\rho z_2\right)^2\right\}, \qquad (3)$$

for $z_1, z_2 \sim \mathcal{N}(0,1)$ independently. Then Eq. (1), (2) and (3) together form the *fundamental equations* for deep random weight-tied autoencoders, in either one of the following senses:

**Interpretation 1.** For $1 \ll \ell \ll L$, in the limit $L \to \infty$ (and $\ell \to \infty$ at a pace sufficiently slow compared to $L$), we have $\bar{\tau}_\ell \to \bar{\tau}$, $\gamma_\ell \to \gamma$ and $\rho_\ell \to \rho$, where $\bar{\tau}$ is a stable fixed point solution to $\bar{\tau}^2 = T(\bar{\tau}^2)$, and $(\gamma, \rho)$ is then jointly a stable fixed point solution to $\gamma = G(\gamma, \rho)$ and $\rho = R(\gamma, \rho)$. In light of Theorem 1, $(\bar{\tau}, \gamma, \rho)$ describes the behavior of an intermediate $\hat{x}_\ell$:

$$\hat{x}_\ell \cong S_{\mathrm{sig}}\sigma(x_{\ell-1}) + S_{\mathrm{var}}z, \quad S_{\mathrm{sig}} = \beta\gamma, \quad S_{\mathrm{var}} = \sqrt{\beta}\rho,$$

where $z \sim \mathcal{N}\left(0, I_{n_{\ell-1}}\right)$ independent of $x_{\ell-1}$, and $x_{\ell-1} \cong \bar{\tau}z'$ for $z' \sim \mathcal{N}\left(0, I_{n_{\ell-1}}\right)$. If the convergences $\bar{\tau}_\ell \to \bar{\tau}$, $\gamma_\ell \to \gamma$ and $\rho_\ell \to \rho$ are fast, then the majority of the intermediate layers are well approximately described by the above, in the regime of large $L$.

**Interpretation 2.** Suppose that for $\bar{\tau}_0 = \|x\|/\sqrt{n_0}$, we impose the constraint $\bar{\tau}_0^2 = \mathbb{E}\left\{\sigma(\bar{\tau}z)^2\right\}$. This should be interpreted as follows: starting with a chosen $\bar{\tau}$, we normalize the input data $x$ according to $\bar{\tau}_0^2 = \mathbb{E}\left\{\sigma(\bar{\tau}z)^2\right\}$; then we choose $\sigma_W^2 \leq \sigma_{W,\max}^2$ and $\sigma_b^2$ according to Eq. (1). Under this constraint, it is easy to see that $\bar{\tau}_\ell = \bar{\tau}$ for all $\ell \geq 1$, and hence the norm of the input to each of the encoder's hidden layers is preserved by Claim (a) of Theorem 1. We then have that as $L \to \infty$, at small $\ell \geq 2$ (i.e., at the layers near the two ends of the autoencoder), $\gamma_\ell \to \gamma$ and $\rho_\ell \to \rho$, where $(\gamma, \rho)$ is jointly a stable fixed point of $\gamma = G(\gamma, \rho)$ and $\rho = R(\gamma, \rho)$. The autoencoder's output is then, in this limit,

$$\hat{x} \cong S_{\mathrm{sig}}x + S_{\mathrm{var}}z, \quad S_{\mathrm{sig}} = \beta\gamma, \quad S_{\mathrm{var}} = \sqrt{\beta}\rho,$$

for $z \sim \mathcal{N}(0, I_{n_0})$ independent of $x$.

In either cases, we see that $\hat{x}_\ell$ or $\hat{x}$ is a composition of a signal component and a Gaussian variation component. Their respective strengths $S_{\mathrm{sig}}$ and $S_{\mathrm{var}}$ admit simple expressions, thanks to the infinite-$L$ simplification[5]. We note that $G$ and $R$ do not take $\sigma_b^2$ as a parameter. We refer to Appendix C.1 for the computation of $\gamma$ and $\rho$. Fig. 1 shows that our asymptotic simplification is quite accurate already for $L$ on the order of a few tens.

We also remark that the equation $\bar{\tau}^2 = T(\bar{\tau}^2)$ is known in the signal propagation analysis of random feedforward networks (Poole et al. (2016)), but the equations $\gamma = G(\gamma, \rho)$ and $\rho = R(\gamma, \rho)$ are new. We also observe that in these equations, there is the presence of $\alpha$ (through $\beta$), which is typically missing from such analyses. Hence unlike untied structures, one may expect to see architectural constraints in analyses of weight-tied structures.

### 3.2 Phase transition of the fixed point

With the infinite depth simplification, one question is on the existence of the solutions to Eq. (2) and (3), and how these fixed points look like. (Eq. (1) is well-studied, cf. Poole et al. (2016); Hayou et al. (2018), and we will not analyze it here.) We note that $\gamma = 0$ is always a solution to Eq. (2), in

---

biases. In addition, it is a common practice to initialize an untrained neural network (hence a random one) with zero biases (Goodfellow et al. (2016)). This assumption also helps making the analysis more tractable. Other assumptions seem restrictive, but can be relaxed without affecting the analysis that follows, and hence are made for ease of presentation. For example, the assumption $\alpha_\ell = \alpha$ for all $\ell$ may be relaxed to $\alpha_\ell = \alpha$ for most $\ell$ such that $1 \ll \ell \ll L$.

[5]Recall we consider the limit of $n \to \infty$ before taking $L \to \infty$. Without a proof, we expect that our results hold for $L = o(\log n)$, so practically this means $1 \ll L \ll \Omega(\log n)$.

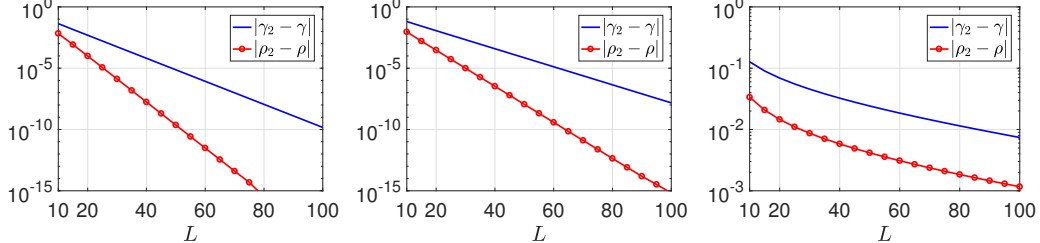

Figure 1: The gaps $|\gamma_2 - \gamma|$ and $|\rho_2 - \rho|$ versus the depth $L$, where $\gamma_2$ and $\rho_2$ (which are dependent on $L$) are as in Section 2.2, and $\gamma$ and $\rho$ (the infinite-$L$ limits of $\gamma_2$ and $\rho_2$) are from Eq. (2) and (3). Here all activations are $\tanh$, $\bar{\tau}^2 = 1.2$, $\sigma_b^2 = 0.211$, $\sigma_W^2 = 2.312 < \sigma_{W,\max}^2 \approx 2.806$, and $\bar{\tau}_0^2 \approx 0.4276$, which satisfies $\bar{\tau}_0^2 = \mathbb{E}\left\{\sigma(\bar{\tau}z)^2\right\}$. From left to right: $\alpha = 0.9$, $\alpha = 1.0$ and $\alpha = 1.5$. The gaps decrease exponentially with the depth $L$.

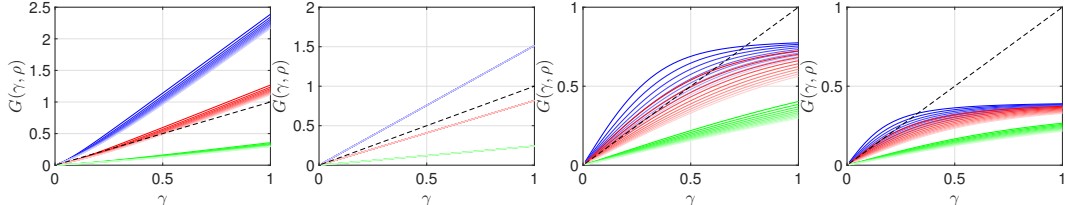

Figure 2: The mapping $\gamma \mapsto G(\gamma, \rho)$ for $\bar{\tau}^2 = 1$ and $\beta = 5$ (blue), $\beta = 2.7$ (red), $\beta = 0.8$ (green). The color intensity varies with $\rho \in [0.1, 1]$ with equal spacings, where the darkest curve corresponds to $\rho = 0.1$, and the lightest is $\rho = 1$. From left to right: $\varphi, \sigma$ are ReLU; $\varphi$ is ReLU, $\sigma$ is $\tanh$; $\varphi, \sigma$ are $\tanh$; $\varphi$ is $\tanh$, $\sigma$ is ReLU. A fixed point is an intersection between this mapping and the identity line (black dashed).

which case Eq. (3) also has a solution, for instance, when $\varphi(0) = 0$ such as ReLU or $\tanh$ (which admits $\rho = 0$). However $\gamma = 0$ is trivial, since it implies $S_{\text{sig}}$ is zero. We will be interested in the existence of non-trivial and stable fixed points. To ease visualization, for the moment, let us consider Eq. (2) only. Fig. 2 shows $\gamma \mapsto G(\gamma, \rho)$ for different $\rho$, $\beta$, $\varphi$ and $\sigma$ for $\bar{\tau}^2 = 1$. For a given $\varphi$ and $\sigma$, depending on $\beta$ and $\rho$, one may observe one or more fixed points, one of which is at $\gamma = 0$ and can be stable or unstable. When $\gamma = 0$ is the only fixed point but is unstable, we have $\gamma = \infty$ as the "stable solution" to Eq. (2). The solution landscape changes drastically with $\beta$; for instance, when $\sigma = \varphi = \tanh$, $\gamma = 0$ is the only and stable fixed point when $\beta$ is small, but it becomes unstable and a new fixed point at $\gamma > 0$ emerges when $\beta$ is sufficiently large. This hints at certain phase transition behaviors as $\beta$ varies.

In Appendix C.2, we perform a detailed analysis of Eq. (2) and (3), supported by several rigorously proven properties. In the following, by an initialization for $\gamma$ and $\rho$, we mean $\gamma_{L+1}$ and $\rho_{L+1}$ as in Section 2.2, and by convergence to $\gamma$ and $\rho$, we mean the convergences as in Section 3.1. We highlight some results from the analysis for specific pairs of $\varphi$ and $\sigma$:

**ReLU $\varphi$ and $\sigma$.** We have two phase transitions at $\beta = 2$ and at $\beta = 4$. When $\beta < 2$, with any initialization, we have convergence to $\gamma = 0$ and $\rho = 0$. When $2 < \beta < 4$, we have, with certain initializations, convergence to $\gamma = 0$ and divergence to $\rho = +\infty$, and with certain other initializations, divergence to $\gamma = +\infty$ and $\rho = +\infty$. These include almost all possible initializations. When $\beta \geq 4$, with any non-zero initialization, we have divergence to $\gamma = +\infty$ and $\rho = +\infty$.

**ReLU $\varphi$ and $\tanh$ $\sigma$.** We have two phase transitions at $\beta = 2$ and $\beta = \beta_0(\bar{\tau}) \in (2, \infty)$. When $\beta < 2$, with any initialization, we have convergence to $\gamma = 0$ and $\rho = 0$. When $2 < \beta < \beta_0$, with any non-zero initialization, we have convergence to $\gamma = 0$ and divergence to $\rho = +\infty$. When $\beta > \beta_0$, with any non-zero initialization, we have divergence to $\gamma = +\infty$ and hence $\rho = +\infty$.

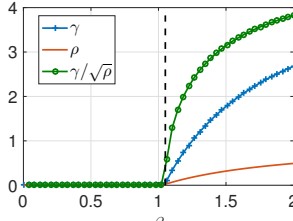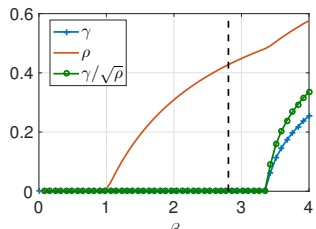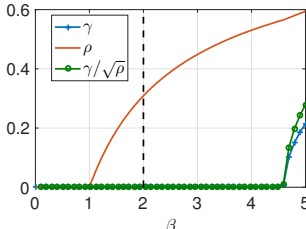

Figure 3: $\gamma$ and $\rho$ versus $\beta$, as solved with Eq. (2) and (3). The vertical dotted line is $\beta = \sigma_{W,\max}^2$. From left to right: (1) $\varphi = \sigma = \tanh$ and $\bar{\tau}^2 = 0.0259$, (2) $\varphi = \sigma = \tanh$ and $\bar{\tau}^2 = 1.2$, and (3) $\varphi = \tanh$, $\sigma$ is the ReLU, and $\bar{\tau}^2 = 0.2$.

**tanh $\varphi$ and $\sigma$.**     We have two phase transitions at $\beta = 1$ and $\beta = \beta_0(\bar{\tau}) > 1$. When $\beta \leq 1$, we have convergence to $\gamma = \rho = 0$. When $1 < \beta < \beta_0$, with any non-zero initialization, we have convergence to $\gamma = 0$ and $\rho \in (0, 1)$. When $\beta > \beta_0$, with any non-zero initialization, we have convergence to $\gamma > 0$ and $\rho \in (0, 1)$. For $\bar{\tau} > 0$, $\gamma$ cannot grow to $+\infty$ as $\beta$ varies. We note that $\beta_0 \to 1$ if $\bar{\tau}^2 \to 0$, and in the case $\alpha = 1$, this implies $\sigma_W^2 \to 1$. With respect to Eq. (1), we then have $\sigma_b^2 \to 0$. An illustration is given in Fig. 3.

**tanh $\varphi$ and ReLU $\sigma$.**     We have a picture similar to the case $\varphi = \sigma = \tanh$, with a crucial difference that one cannot have $\beta_0$ be close to 1. An illustration is given in Fig. 3.

$\gamma$ and $\rho$ thus exhibit phase transitions, depend crucially on the choice of activations (especially the decoder activation $\varphi$), and can be trivialized (i.e., being zero or infinity) as in the case of ReLU $\varphi$ and $\sigma$. It is remarkable that the above pictures are general for many other activations, as suggested by our analysis. In the next sections, we explore the implications of these behaviors.

### 3.3 APPROXIMATE INFERENCE AT INFINITE DEPTH

Theorem 1 gives a quantitatively exact sense of how the random weight-tied autoencoder performs "approximate inference". Here we will be interested in stronger notions. A first question is: does it explain reversibility? Reversibility, as mathematically formalized in previous works Arora et al. (2015); Gilbert et al. (2017), quantitatively concerns with how small the quantity $\mathcal{E} = \|\boldsymbol{x} - \hat{\boldsymbol{x}}\|^2 / n_0$ is. The smaller it is, the better the decoder "reverses" the encoder. This formalized notion is an attempt to give a theoretical understanding of empirical findings that the input could be reproduced from the values of hidden layers of a trained feedforward network. Let us now consider the infinite depth simplification under Interpretation 2 of Section 3.1. We have $\mathcal{E} \simeq (S_{\text{sig}} - 1)^2 \bar{\tau}_0^2 + S_{\text{var}}^2$. As such, for $\mathcal{E} \approx 0$ with high probability, one must have $S_{\text{sig}} \approx 1$ and $S_{\text{var}} \approx 0$, hence $\rho \approx 0$. Consequently, $\mathbb{E}\left\{\varphi\left(\sigma\left(\bar{\tau}z_1\right)\right)^2\right\} \approx 0$. For $\bar{\tau} > 0$, $\mathbb{E}\left\{\varphi\left(\sigma\left(\bar{\tau}z_1\right)\right)^2\right\} = 0$ is impossible for any non-trivial activations (unless the activation outputs zero almost everywhere). Strikingly, in light of Section 3.2, when $\varphi$ and $\sigma$ are both ReLU, we have that $S_{\text{sig}}$ and $S_{\text{var}}$ are either 0 or $+\infty$, in which case $\mathcal{E} \geq \bar{\tau}_0^2$ and can become unbounded. While this does not contradict the results in Arora et al. (2015) (which also concerns with ReLU activations, but with specific choices of the biases and limited depth, and hence is in a different setting), our discussion suggests that random weight-tied models may be insufficient to explain reversibility.

A second question is: does the model perform signal recovery? In this case, we are interested in whether $\hat{\boldsymbol{x}} \cong c\boldsymbol{x}$ for some constant $c$ not necessarily 1. Similar to the above, this requires $S_{\text{var}} = 0$, hence $\rho = 0$, and $\mathbb{E}\left\{\varphi\left(\beta\gamma\sigma\left(\bar{\tau}z_1\right)\right)^2\right\} = 0$. For non-trivial $\varphi$ and $\sigma$, this requires $\varphi(0) = 0$ and $\gamma = 0$. Many activations do not conform with the former, and the latter implies $\hat{\boldsymbol{x}} \cong 0$ undesirably. This provides a negative answer to the question.

A critic may argue that in expectation, $\mathbb{E}\{\hat{\boldsymbol{x}}\} \approx S_{\text{sig}}\boldsymbol{x}$, and as per Section 3.2, there are cases where $\gamma > 0$ and hence $S_{\text{sig}} > 0$. Yet in fact, this in-expectation property can already be observed in the simple setting of linear shallow autoencoders (Arora et al. (2015)). What is ignored in such argument is that in many cases, $S_{\text{var}} > 0$ whenever $S_{\text{sig}} > 0$, in light Section 3.2. Our analysis hence mitigates

the shortcoming of the in-expectation approach, and gives a more precise understanding of what the random weight-tied autoencoder can and cannot achieve when the depth becomes large.

## 3.4 COMPARISON WITH THE SHALLOW CASE

Our result also allows for the case of $L = 1$ a shallow autoencoder. In particular, taking a parallel setting with Section 3.1 (in particular, Interpretation 2), by Theorem 1,

$$\hat{\boldsymbol{x}} \cong S_{\text{sig}} \boldsymbol{x} + S_{\text{var}} \boldsymbol{z}, \quad S_{\text{sig}} = \beta\gamma, \quad S_{\text{var}} = \sqrt{\beta}\rho,$$

in which, with $\varphi_{\text{hid}}$ being the activation in the hidden layer,

$$\gamma = \frac{1}{\bar{\tau}^2} \mathbb{E}\left\{ \bar{\tau} z \varphi_{\text{hid}}\left(\bar{\tau} z\right)\right\}, \quad \rho = \mathbb{E}\left\{\varphi_{\text{hid}}\left(\bar{\tau} z\right)^2\right\}.$$

Some observations follow. In the shallow case, $\gamma > 0$ and $\rho > 0$ and both are bounded regardless of the parameters, except for trivial edge cases such as $\bar{\tau}^2 = 0$ or $\varphi_{\text{hid}}\left(\cdot\right) = 0$. Furthermore, $\gamma$ and $\rho$ are independent of $\beta$, for a fixed $\bar{\tau}$. As such, there is no phase transition in $\gamma$ and $\rho$ as $\beta$ changes. We also have $S_{\text{sig}}\left(\beta\right) \propto \beta$ and $S_{\text{var}}\left(\beta\right) \propto \sqrt{\beta}$. Hence, the signal component dominates with $S_{\text{sig}}/S_{\text{var}} \propto \sqrt{\beta}$. Again this happens regardless of parameter choices.

In comparison with the infinite depth case, for $\varphi = \sigma = \tanh$ or $\varphi = \tanh$ and $\sigma$ being the ReLU, as observed from Fig. 3, in certain regimes, $\gamma$, $\rho$ and $\gamma/\sqrt{\rho}$ can grow (sublinearly) with $\beta$, and hence $S_{\text{sig}}\left(\beta\right) = \Omega\left(\beta\right)$ [6], $S_{\text{var}}\left(\beta\right) = \Omega\left(\sqrt{\beta}\right)$, and the signal component dominates with $S_{\text{sig}}/S_{\text{var}} = \Omega\left(\sqrt{\beta}\right)$. In particular, near the phase transition of $\gamma$, $S_{\text{sig}}/S_{\text{var}} = \Omega\left(\beta^{1.5}\right)$. Recalling that $\beta = \alpha\sigma_W^2$, this implies for the infinite depth case, as compared to the shallow one, firstly a slight perturbation in $\sigma_W^2$ may result in a larger perturbation in the signal's strength, and secondly an architecture using larger $\alpha$ may gain more in terms of amplification of the signals. In short, the deep autoencoder is more sensitive to slight changes in the parameters. As evident in Section 3.2, the case $\varphi$ being the ReLU also exhibits extreme sensitivity, in that it is possible for a slight perturbation in $\beta$ to drastically change $\gamma$ and $\rho$. As suggested by Fig. 1, it should be the case already for $L$ about a few tens. It is, however, at the expense of much care in the selection of parameters, since there are continuous regimes in which the infinite depth diminishes $S_{\text{sig}}$ and $S_{\text{var}}$ to zero or boost them to infinity, a situation that never occurs in the shallow case.

*Remark* 4. Sensitivity to perturbations is implied by expressivity, a notion put forth in Poole et al. (2016) in the study of random feedforward networks. Hence we expect that sensitivity is a common feature of various types of deep neural networks.

## 3.5 IMPLICATIONS TO TRAINING INITIALIZATION

We examine the implications of Interpretation 1 in Section 3.1 to trainability of the weight-tied autoencoder. We first state our hypothesis, then test it with experiments.

**The hypothesis.** Since the majority of intermediate layers can be described approximately by $\gamma$ and $\rho$ (as well as $\bar{\tau}$) and the random weight-tied autoencoder is in fact one at initialization, appropriate values of $\gamma$, $\rho$ and $\bar{\tau}$ (by a suitable choice of $\sigma_W^2$ and $\sigma_b^2$) should lead to better trainability. In particular, if one of them is $\infty$, we expect numerical errors or too large values resulting in quick saturation, both of which render the autoencoder untrainable. If $\gamma = \rho = 0$ in a neighborhood of the chosen $\sigma_W^2$ and $\sigma_b^2$, we expect that the progress is slowed down in the beginning. If such pitfalls are avoided, the autoencoder is expected to show a faster progress.

Our analysis in Section 3.2 shows that those pitfalls can occur for a wide range of parameters. If the hypothesis is true, $\sigma_W^2$ and $\sigma_b^2$ must then be chosen carefully when $L$ is large. As a special remark on the case $\varphi$ is the ReLU, when $\alpha = 1$, the hypothesis suggests taking $\sigma_W^2 = 2$ and $\sigma_b^2 = 0$. This coincides with the celebrated He initialization (He et al. (2015)), which however considers feedforward networks only. Interestingly this does require $\sigma$ to be the ReLU; for instance, $\sigma$ can be $\tanh$, in which case the argument in (He et al. (2015)) is not applicable.

We shall also examine edge of chaos (EOC) initializations (Schoenholz et al. (2016); Pennington et al. (2017)) (see Appendix E.1). The EOC initialization enables better signal propagation in deep feedforward networks, and in our context, is relevant to the encoder part with the activation $\sigma$.

---

[6]By $f(\beta) = \Omega(g(\beta))$ in a certain range of $\beta$, we mean $\frac{\mathrm{d}}{\mathrm{d}\beta}[f(\beta)/g(\beta)] \geq 0$ on this range.

| No. | $\varphi$ | $\sigma$ | $\sigma_W^2$ | $\sigma_b^2$ | $\bar{\tau}^2$ | EOC | Trainable | Slowed | Inf |
|---|---|---|---|---|---|---|---|---|---|
| 1 | ReLU | ReLU | 2.0 | 0.0 | − | x | x | | |
| 2 | ReLU | ReLU | 1.0 | 0.1 | 0.2 | | | x | |
| 3 | ReLU | ReLU | 2.5 | 0.0 | $\infty$ | | | | x |
| 4 | ReLU | tanh | 2.0 | 0.0 | 0.618 | | x | | |
| 5 | ReLU | tanh | 1.05 | $2.01 \times 10^{-5}$ | 0.0259 | xx | | x | |
| 6 | ReLU | tanh | 2.505 | 0.3 | 1.460 | | | | x |
| 7 | tanh | tanh | 1.05 | $2.01 \times 10^{-5}$ | 0.0259 | xx | x | | |
| 8 | tanh | tanh | 0.5 | 0.0136 | 0.0259 | | | x | |
| 9 | tanh | tanh | 2.312 | 0.211 | 1.2 | x | x | | |
| 10 | tanh | tanh | 0.5 | 0.986 | 1.2 | | | x | |
| 11 | tanh | tanh | 1.0 | 0.771 | 1.2 | | x | | |
| 12 | tanh | ReLU | 2.0 | 0.0 | − | x | x | | |
| 13 | tanh | ReLU | 1.0 | 0.1 | 0.2 | | x | | |
| 14 | tanh | ReLU | 0.5 | 0.15 | 0.2 | | | x | |

Table 1: List of initialization schemes for each pair of $\varphi$ and $\sigma$, for $\alpha = 1$. Here "−" indicates a positive finite value that depends on the choice of $\varphi_L$ (for which we choose the ReLU), but its exact value is irrelevant for our purpose. "EOC" indicates whether the scheme is an EOC initialization with respect to $\sigma$, and "xx" indicates an EOC scheme that is found to be the better one among all EOC initializations with Gaussian weights (Pennington et al. (2017)). "Trainable" indicates better trainability in the beginning as predicted by our theory. "Slowed" indicates $\gamma = \rho = 0$ in a neighborhood. "Inf" indicates either $\gamma \to \infty$ or $\rho \to \infty$. The schemes with $\varphi = \tanh$ should be reflected against Fig. 3.

**Experiments.** Table 1 lists several initialization schemes with $\alpha = 1$, which are chosen such that the hypothesis can be tested separately for each pair $\varphi$ and $\sigma$. We perform simple experiments on a weight-tied vanilla autoencoder as described in Section 3.1: $L = 100$, all hidden dimensions of $400$, identity input activation $\sigma_0$, and decoder biases initialized to zero. This sets $\alpha_\ell = \alpha = 1$ for $\ell \geq 2$; here $\alpha_1 \neq 1$ is irrelevant in light of Interpretation 1. We train the autoencoder on the MNIST dataset with mini-batch gradient descent with a batch size of 250 and without regularizations, for $5 \times 10^5$ iterations (equivalent to 2500 epochs). We perform the experiments in two settings:

- **Setting 1:** The output activation $\varphi_0$ is $\tanh$, MNIST images are normalized to $[-1, +1]$, and the learning rate is fixed at $5 \times 10^{-3}$. This is standard for MNIST.

- **Setting 2:** $\varphi_0$ is the identity, MNIST images are unnormalized (i.e., normalized to $[0, +1]$), and the learning rate is fixed at $3 \times 10^{-3}$. This is common for regression.

These learning rates are chosen so that the learning dynamics is typically smooth, in light of recent works Mei et al. (2018); Smith & Le (2018). We use the normalized $\ell_2^2$ loss $\|\hat{x} - x\|^2 / \|x\|^2$, and are primarily interested in this loss as a quality measure, since we only focus on trainability[7]. We also do not apply techniques such as greedy layer-wise pre-training, drop-out or batch normalization. The results are plotted in Fig. 4. See also Appendix D.1 for visualization of the reconstructions, and Appendix D.2 for the evolution over a broader range of parameters. Note that we plot the evolution in the logarithmic scale of time, since it is typically smooth and revealing on this scale, as found in prior works Baity-Jesi et al. (2018); Mei et al. (2018) and also evident from the plots.

**Discussion.** The results are in good agreement with our hypothesis. (Recall we test the hypothesis separately for each pair $\varphi$ and $\sigma$, for which the involved schemes share the same architecture and only differ in the initialization.) Note that as predicted, in Setting 2, Scheme 3 and 6 are trapped with numerical errors, and in Setting 1, they saturate quickly at a high loss. As such, we do not include the results of Scheme 3 and 6 in Fig. 4.

---

[7]The chosen loss is slightly different from the traditional $\ell_2^2$ loss $\|\hat{x} - x\|^2$. On one hand, we found from our experiments that these two losses perform comparably, with the normalized loss typically yielding slight improvements, provided that the learning rates are scaled appropriately. On the other hand, the normalized loss allows ease for interpretation.

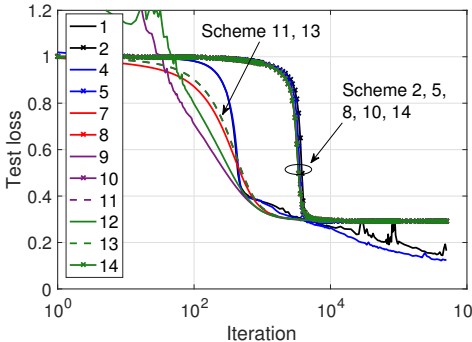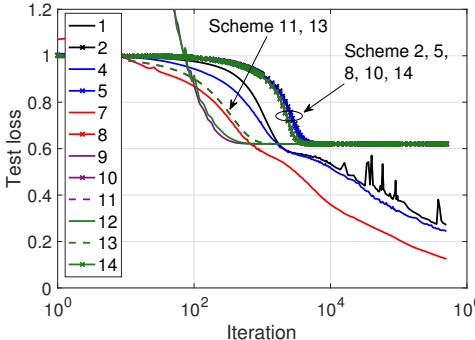

Figure 4: Test loss $\|\hat{x} - x\|^2 / \|x\|^2$ of the schemes from Table 1. Left: the setting with $\varphi_0 = \tanh$ (Setting 1). Right: the setting where $\varphi_0$ is the identity (Setting 2).

We see from the figure that Scheme 2, 5, 8, 10 and 14 show much slower progresses, by a factor of 3 to 10 times in terms of training iterations to reach the same loss. Hence a good amount of training time can be saved by an appropriate initialization. Interestingly Scheme 5 is in fact a special EOC initialization that Pennington et al. (2017) found to be the better one among all EOC schemes with Gaussian weights for $\tanh$ activation. This last observation shows that having good signal propagation through the encoder is far from being a sufficient condition for trainability.

Among the schemes, only Scheme 1 and 4 in Setting 1 and only Scheme 1, 4 and 7 in Setting 2 have their eventual trained networks produce meaningful reconstructions, whereas the rest always output some "average" of the training set regardless of the input, at the end of $5 \times 10^5$ iterations (see Appendix D.1). It is unclear whether this is a bad local minimum, or whether these schemes take much longer to show further progresses. An explanation is beyond our current theory, and it is an open question how to create a scheme with meaningful trainability. Remarkably all the schemes that show slower initial progresses (Scheme 2, 5, 8, 10 and 14) are among those that could not yield meaningful reconstructions.

We observe that in Setting 2, the $\tanh$ network under Scheme 7 is best performing in terms of the reconstruction loss, and its progress does not seem to reach a plateau after $5 \times 10^5$ iterations. In both settings, Scheme 4, which is a hybrid of ReLU and $\tanh$ activations, shows slight improvements over Scheme 1, which is a purely ReLU network. This extends the conclusion in Pennington et al. (2017) to the context of weight-tied autoencoders: reasonable training at a large depth is possible even for the notoriously difficult $\tanh$ activation, and this necessarily requires careful initializations.

Overall we see that our experiments confirm the hypothesis, showing an intimate connection between the phase transition behaviors found by our theory and trainability of the autoencoders.

## 4 DISCUSSION

This paper has shown quantitative answers to the three questions posed in Section 1. This feat is enabled by an exact analysis via Theorem 1. The theorem is stated in a general setting, allowing varying activations, weight variances, etc, but our analyses in Section 3 have made several simplifications. This leaves a question of whether these simplifications can be relaxed, and how the picture changes accordingly, for instance, when the parameters vary across layers, similar to Yang & Schoenholz (2018). Many other questions also remain. For example, what would be the covariance structure between the outputs of two distinct inputs? How does the network's Jacobian matrix look like? These questions have been answered in the feedforward case (Poole et al. (2016); Pennington et al. (2017)), but we believe answering them is more technically involved in our case. We have also seen that an autoencoder that shows initial progress may not necessarily produce meaningful reconstruction eventually after training, and hence much more work is needed to understand the training dynamics far beyond initialization. Recent works Mei et al. (2018); Rotskoff & Vanden-Eijnden (2018); Sirignano & Spiliopoulos (2018); Chizat & Bach (2018) have made progresses in this direction for shallow networks.

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

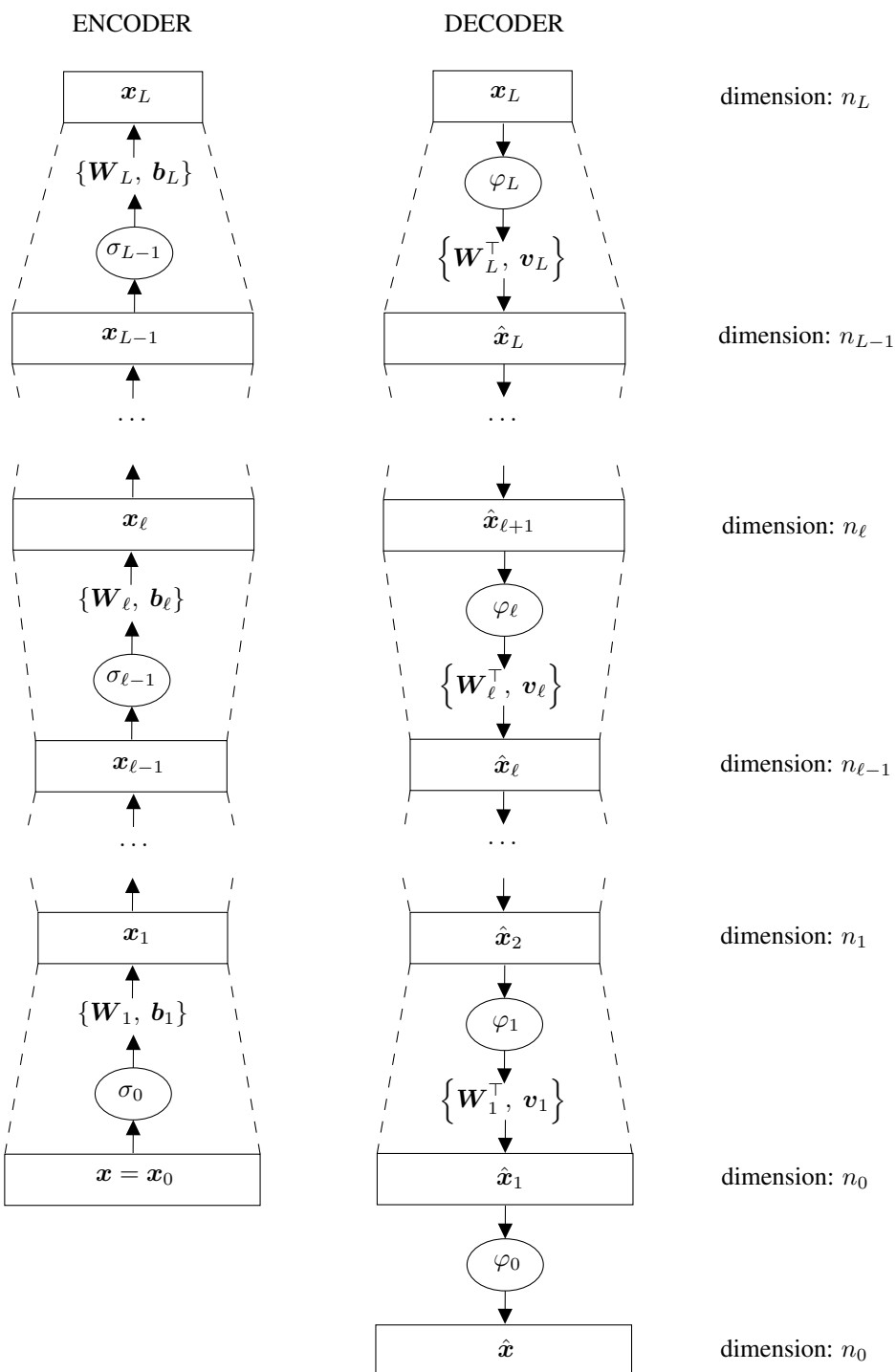

Figure 5: Schematic diagram of the weight-tied autoencoder as described in Section 2.1. Left: the encoder. Center: the decoder. Right: dimensions of the corresponding vectors.

# A    PROOF OF THEOREM 1

In the following, we give an outline of the proof of Theorem 1, and the complete proof. First, we start with a few notations and definitions.

## A.1    DEFINITIONS AND NOTATIONS

We recall the setting in Section 2.1 (see also Fig. 5 for a schematic diagram of the autoencoder). We define $\{g_\ell : \mathbb{R}^{n_{\ell-1}} \mapsto \mathbb{R}^{n_{\ell-1}}\}_{\ell=1,...,L}$ inductively as follows:

$$g_L(\boldsymbol{u}) = \boldsymbol{W}_L^\top \varphi_L(\boldsymbol{W}_L \sigma_{L-1}(\boldsymbol{u}) + \boldsymbol{b}_L) + \boldsymbol{v}_L,$$

$$g_\ell(\boldsymbol{u}) = \boldsymbol{W}_\ell^\top \varphi_\ell(g_{\ell+1}(\boldsymbol{W}_\ell \sigma_{\ell-1}(\boldsymbol{u}) + \boldsymbol{b}_\ell)) + \boldsymbol{v}_\ell, \quad \ell = L-1, ..., 1.$$

It is easy to see that $\hat{\boldsymbol{x}}_\ell = g_\ell(\boldsymbol{x}_{\ell-1})$ for $\ell = 1, ..., L$. Essentially $g_\ell$'s represent the autoencoding mappings computed by the inner layers.

We use bold-face letters (e.g. $\boldsymbol{x}$, $\boldsymbol{W}$) to denote vectors or matrices. We use $C$ throughout to denote an arbitrary (and immaterial) constant that is independent of the dimensions. For two vectors $\boldsymbol{x}$ and $\boldsymbol{y}$, $\langle \boldsymbol{x}, \boldsymbol{y} \rangle$ denotes their inner product. We use $\boldsymbol{I}$ for the identity matrix, and $\boldsymbol{I}_n$ to emphasize its dimensions $n \times n$. We use $\|\cdot\|$ to denote the usual Euclidean norm, $\|\cdot\|_\infty$ the infinity norm, and $\|\boldsymbol{M}\|_2$ the maximum singular value of a matrix $\boldsymbol{M}$. For a sigma-algebra $\mathcal{F}$ and two random variables $X$ and $Y$, $X|\mathcal{F} \overset{\mathrm{d}}{=} Y$ means that for any integrable function $\phi$ and any $\mathcal{F}$-measurable bounded random variable $Z$, $\mathbb{E}\{\phi(X) Z\} = \mathbb{E}\{\phi(Y) Z\}$. We write $X \overset{\mathrm{d}}{=} Y$ when $\mathcal{F}$ is the trivial sigma-algebra.

A sequence of functions $\phi_n : \mathbb{R}^n \mapsto \mathbb{R}$ is uniformly pseudo-Lipschitz if there exists a constant $C$, independent of $n$, such that for any $\boldsymbol{x}, \boldsymbol{y} \in \mathbb{R}^n$,

$$|\phi_n(\boldsymbol{x}) - \phi_n(\boldsymbol{y})| \leq C \left(1 + \frac{\|\boldsymbol{x}\|}{\sqrt{n}} + \frac{\|\boldsymbol{y}\|}{\sqrt{n}}\right) \frac{\|\boldsymbol{x} - \boldsymbol{y}\|}{\sqrt{n}}.$$

A sequence of functions $\phi_n : \mathbb{R}^n \mapsto \mathbb{R}^n$ is uniformly Lipschitz if there exists a constant $C$, independent of $n$, such that for any $\boldsymbol{x}, \boldsymbol{y} \in \mathbb{R}^n$, $\|\phi_n(\boldsymbol{x}) - \phi_n(\boldsymbol{y})\| \leq C \|\boldsymbol{x} - \boldsymbol{y}\|$. These definitions are adopted from Berthier et al. (2017). For two sequences of random variables $X_n \in \mathbb{R}$ and $Y_n \in \mathbb{R}$ indexed by $n$, we write $X_n \simeq Y_n$ to mean that $X_n - Y_n \to 0$ in probability. The same meaning holds when $Y_n$ is deterministic. For two sequences of random vectors $\boldsymbol{X}_n \in \mathbb{R}^n$ and $\boldsymbol{Y}_n \in \mathbb{R}^n$ indexed by $n$, we write $\boldsymbol{X}_n \cong \boldsymbol{Y}_n$ if for any sequences of uniformly pseudo-Lipschitz test functions $\phi_n : \mathbb{R}^n \mapsto \mathbb{R}$, $\phi_n(\boldsymbol{X}_n) \simeq \mathbb{E}\{\phi_n(\boldsymbol{Y}_n)\}$ (and hence in this context, we do not need $\boldsymbol{X}_n$ and $\boldsymbol{Y}_n$ to be defined on a joint probability space).

## A.2    OUTLINE OF THE PROOF OF THEOREM 1

We state several results that are key to prove Theorem 1.

**Proposition 5.** *Consider the asymptotic setting $n \to \infty$, with some sequence $m = m(n)$ such that $m/n \to \alpha > 0$ as $n \to \infty$. Let $\varphi : \mathbb{R} \mapsto \mathbb{R}$ and $\sigma : \mathbb{R} \mapsto \mathbb{R}$ be Lipschitz continuous scalar functions. Consider a sequence of uniformly Lipschitz functions $g : \mathbb{R}^m \mapsto \mathbb{R}^m$, and sequences of vectors $\boldsymbol{b} \in \mathbb{R}^m$, $\boldsymbol{v} \in \mathbb{R}^n$, $\boldsymbol{u} \in \mathbb{R}^n$ such that*

$$\frac{\|\boldsymbol{b}\|^2}{m} \leq C, \quad 0 < c \leq \frac{\|\sigma(\boldsymbol{u})\|^2}{n} \leq C, \quad \frac{\|\boldsymbol{v}\|^2}{n} \leq C$$

*for all sufficiently large $n$. Let us define*

$$f(\boldsymbol{u}) = \boldsymbol{W}^\top \varphi(g(\boldsymbol{W}\sigma(\boldsymbol{u}) + \boldsymbol{b})) + \boldsymbol{v},$$

*where $\boldsymbol{W} \in \mathbb{R}^{m \times n}$ with $W_{ij} \sim \mathcal{N}(0, \sigma_W^2/n)$ i.i.d. for $\sigma_W > 0$. Let*

$$\gamma = \mathbb{E}_{\tilde{\boldsymbol{z}}}\left\{\frac{1}{m\tau^2}\langle \tau\tilde{\boldsymbol{z}}, \varphi(g(\tau\tilde{\boldsymbol{z}} + \boldsymbol{b}))\rangle\right\}, \quad \rho = \mathbb{E}_{\tilde{\boldsymbol{z}}}\left\{\frac{1}{m}\|\varphi(g(\tau\tilde{\boldsymbol{z}} + \boldsymbol{b}))\|^2\right\}, \quad \tau^2 = \sigma_W^2 \frac{\|\sigma(\boldsymbol{u})\|^2}{n},$$

*for $\tilde{\boldsymbol{z}} \sim \mathcal{N}(0, \boldsymbol{I}_m)$. Then:*

$$f(\boldsymbol{u}) \cong \alpha\sigma_W^2 \gamma \sigma(\boldsymbol{u}) + \sqrt{\alpha\sigma_W^2 \rho}\,\boldsymbol{z} + \boldsymbol{v},$$

*where $\boldsymbol{z} \sim \mathcal{N}(0, \boldsymbol{I}_n)$ (independent of $\boldsymbol{u}$, $\boldsymbol{b}$ and $\boldsymbol{v}$). Here the randomness is solely due to $\boldsymbol{W}$.*

A corollary follows from this proposition.

**Corollary 6.** *Consider the same setting as in Proposition 5. Further assume that $\boldsymbol{b} \sim \mathcal{N}\left(0, \sigma_b^2 \boldsymbol{I}_m\right)$, $\boldsymbol{v} \sim \mathcal{N}\left(0, \sigma_v^2 \boldsymbol{I}_n\right)$ and $\boldsymbol{u} \sim \mathcal{N}\left(0, \sigma_u^2 \boldsymbol{I}_n\right)$, independent of each other and of $\boldsymbol{W}$. Then:*

$$f\left(\boldsymbol{u}\right) \cong \alpha \sigma_W^2 \gamma \sigma\left(\sigma_u \boldsymbol{z}\right) + \sqrt{\alpha \sigma_W^2 \rho + \sigma_v^2} \boldsymbol{z}',$$

*for $\boldsymbol{z}, \boldsymbol{z}' \sim \mathcal{N}\left(0, \boldsymbol{I}_n\right)$ independently. In particular, for $\phi: \mathbb{R} \mapsto \mathbb{R}$ Lipschitz continuous,*

$$\frac{1}{n}\left\|\phi\left(f\left(\boldsymbol{u}\right)\right)\right\|^2 \simeq \mathbb{E}_{z_1, z_2}\left\{\phi\left(\alpha \sigma_W^2 \gamma \sigma\left(\sigma_u z_1\right) + \sqrt{\alpha \sigma_W^2 \rho + \sigma_v^2} z_2\right)^2\right\}, \tag{4}$$

$$\frac{1}{n \sigma_u^2}\left\langle \boldsymbol{u}, \phi\left(f\left(\boldsymbol{u}\right)\right)\right\rangle \simeq \mathbb{E}_{z_1, z_2}\left\{\frac{1}{\sigma_u} z_1 \phi\left(\alpha \sigma_W^2 \gamma \sigma\left(\sigma_u z_1\right) + \sqrt{\alpha \sigma_W^2 \rho + \sigma_v^2} z_2\right)\right\}, \tag{5}$$

*in which*

$$\gamma = \mathbb{E}_{\tilde{\boldsymbol{z}}}\left\{\frac{1}{m \bar{\tau}^2}\left\langle \bar{\tau}\tilde{\boldsymbol{z}}, \varphi\left(g\left(\bar{\tau}\tilde{\boldsymbol{z}}\right)\right)\right\rangle\right\}, \ \rho = \mathbb{E}_{\tilde{\boldsymbol{z}}}\left\{\frac{1}{m}\left\|\varphi\left(g\left(\bar{\tau}\tilde{\boldsymbol{z}}\right)\right)\right\|^2\right\}, \ \bar{\tau}^2 = \sigma_W^2 \mathbb{E}_{z_1}\left\{\sigma\left(\sigma_u z_1\right)^2\right\} + \sigma_b^2,$$

*where $\tilde{\boldsymbol{z}} \sim \mathcal{N}\left(0, \boldsymbol{I}_m\right)$, and $z_1, z_2$ are independently distributed as $\mathcal{N}\left(0, 1\right)$.*

In a nutshell, the proposition and the corollary consider a random weight-tied "autoencoder" with a single hidden layer, with a mapping $g$ in the middle. Since $g$ is not a separable function (i.e., $g$ does not apply entry-wise), this is different from the usual shallow autoencoder, a case that has been investigated in Pennington & Worah (2017); Louart et al. (2018) with techniques and objectives different from ours. By understanding this structure, one can understand the random weight-tied multi-layer autoencoder. Indeed, for each $\ell$, the proposition applies with $\boldsymbol{u}$ being $\boldsymbol{x}_{\ell-1}$, $f\left(\boldsymbol{u}\right)$ being $\hat{\boldsymbol{x}}_\ell$ and $g$ being $g_{\ell+1}$. In other words, this studies the mapping $g_\ell$. One can start with $g_L$, then progressively move to outer layers $g_{L-1}$, $g_{L-2}$, etc, and hence analyze the autoencoder completely by repeating the same procedure $L$ times. We note that in doing so, at each step, one requires certain information about the inner layers to perform calculations for the outer layers, and this is worked out in Corollary 6 via a simple recursive relation. In particular, the left-hand sides of Eq. (4) and Eq. (5) play the role of $\rho$ and $\gamma$ of the outer layer, whereas their right-hand sides involve $\rho$ and $\gamma$ from the inner layers. It is also important to note that the assumption that the weights at different layers are independent is crucial in making $\boldsymbol{u}$, $g$ and $\left(\boldsymbol{W}, \boldsymbol{b}, \boldsymbol{v}\right)$ to be independent, allowing the proposition to be applicable at all steps. This is the idea behind the proof of Theorem 1.

We quickly mention the key proof technique for Proposition 5. The main technical challenge in working with the weight-tied structure $\boldsymbol{W}^\top h\left(\boldsymbol{W}\boldsymbol{u}\right)$, for some $h: \mathbb{R}^m \mapsto \mathbb{R}^m$ and $\boldsymbol{W} \in \mathbb{R}^{m \times n}$ Gaussian, is that whereas $\boldsymbol{y} = \boldsymbol{W}\boldsymbol{u}$ is Gaussian with zero mean thanks to independence between $\boldsymbol{u}$ and $\boldsymbol{W}$, $\boldsymbol{W}^\top h\left(\boldsymbol{y}\right)$ is not since $\boldsymbol{y}$ is correlated with $\boldsymbol{W}$. It is observed in Bolthausen (2014) that conditioning on a linear constraint $\boldsymbol{y} = \boldsymbol{W}\boldsymbol{u}$ (or the sigma-algebra $\mathcal{F}$ generated by $\boldsymbol{y}$ and $\boldsymbol{u}$), one has that $\boldsymbol{W}$ is distributed as a conditional projection component plus an independent Gaussian component:

$$\boldsymbol{W}|\mathcal{F} \overset{\mathrm{d}}{=} \mathbb{E}\left\{\boldsymbol{W}|\mathcal{F}\right\} + \tilde{\boldsymbol{W}} P_{\boldsymbol{u}}^\perp,$$

where $P_{\boldsymbol{u}}^\perp$ is an appropriate projection. Here $\mathbb{E}\left\{\boldsymbol{W}|\mathcal{F}\right\}$ is what propagates the information about $\boldsymbol{u}$. In addition, $\tilde{\boldsymbol{W}}$ is Gaussian and independent of $\mathcal{F}$, and hence exact calculations can be then worked out. We note that the assumption $\boldsymbol{W}$ is Gaussian is crucial for this identity to hold.

### A.3 PROOF OF PROPOSITION 5

We state the Gaussian Poincaré inequality, which will be used multiple times throughout the proof. We remark that the use of the Gaussian Poincaré inequality is unlikely to lead to a tight non-asymptotic result, but is sufficient for our asymptotic analysis.

**Theorem 7** (Gaussian Poincaré inequality). *For $\boldsymbol{z} \sim \mathcal{N}\left(0, \sigma^2 \boldsymbol{I}_n\right)$ and $\phi: \mathbb{R}^n \mapsto \mathbb{R}$ continuous and weakly differentiable, there exists a universal constant $C$ such that $\mathrm{Var}\left\{\phi\left(\boldsymbol{z}\right)\right\} \leq C\sigma^2 \mathbb{E}\left\{\left\|\nabla\phi\left(\boldsymbol{z}\right)\right\|^2\right\}$.*

Now we are ready for the proof.

**Step 1.** We perform Gaussian conditioning. Let $\boldsymbol{y} = \boldsymbol{W}\sigma(\boldsymbol{u})$. Let $\mathcal{F}$ be the sigma-algebra generated by $\boldsymbol{y}$ and $\boldsymbol{u}$. Conditioning on $\mathcal{F}$ is equivalent to conditioning on the linear constraint $\boldsymbol{y} = \boldsymbol{W}\sigma(\boldsymbol{u})$. Following Bayati & Montanari (2011), we have

$$\boldsymbol{W}|\mathcal{F} \stackrel{\mathrm{d}}{=} \boldsymbol{W}P_{\sigma(\boldsymbol{u})} + \tilde{\boldsymbol{W}}P_{\sigma(\boldsymbol{u})}^{\perp},$$

where $\tilde{\boldsymbol{W}} \stackrel{\mathrm{d}}{=} \boldsymbol{W}$ and is independent of $\mathcal{F}$, $P_{\sigma(\boldsymbol{u})} = \sigma(\boldsymbol{u})\sigma(\boldsymbol{u})^{\top}/\|\sigma(\boldsymbol{u})\|^2$ the projection onto $\sigma(\boldsymbol{u})$, and $P_{\sigma(\boldsymbol{u})}^{\perp} = \boldsymbol{I} - P_{\sigma(\boldsymbol{u})}$ the corresponding orthogonal projection. As such, since $\varphi(g(\boldsymbol{y}+\boldsymbol{b}))$ is $\mathcal{F}$-measurable,

$$f(\boldsymbol{u})|\mathcal{F} \stackrel{\mathrm{d}}{=} \frac{1}{\|\sigma(\boldsymbol{u})\|^2}\langle \boldsymbol{W}\sigma(\boldsymbol{u}), \varphi(g(\boldsymbol{y}+\boldsymbol{b}))\rangle\sigma(\boldsymbol{u}) + P_{\sigma(\boldsymbol{u})}^{\perp}\tilde{\boldsymbol{W}}^{\top}\varphi(g(\boldsymbol{y}+\boldsymbol{b})) + \boldsymbol{v}$$

$$= \frac{1}{\|\sigma(\boldsymbol{u})\|^2}\langle \boldsymbol{y}, \varphi(g(\boldsymbol{y}+\boldsymbol{b}))\rangle\sigma(\boldsymbol{u}) + P_{\sigma(\boldsymbol{u})}^{\perp}\tilde{\boldsymbol{W}}^{\top}\varphi(g(\boldsymbol{y}+\boldsymbol{b})) + \boldsymbol{v}.$$

For a sequence of uniformly pseudo-Lipschitz functions $\phi_n : \mathbb{R}^n \mapsto \mathbb{R}$:

$$\phi_n(f(\boldsymbol{u}))|\mathcal{F} \stackrel{\mathrm{d}}{=} \phi_n\left(\frac{\langle \boldsymbol{y}, \varphi(g(\boldsymbol{y}+\boldsymbol{b}))\rangle}{\|\sigma(\boldsymbol{u})\|^2}\sigma(\boldsymbol{u}) + P_{\sigma(\boldsymbol{u})}^{\perp}\tilde{\boldsymbol{W}}^{\top}\varphi(g(\boldsymbol{y}+\boldsymbol{b})) + \boldsymbol{v}\right) \equiv \Phi_n. \quad (6)$$

Up to this point, there is no need for the asymptotics $n \to \infty$. The rest of the proof focuses on $\Phi_n$.

**Step 2.** We show that $\tau$, $\gamma$ and $\rho$ are uniformly bounded as $n \to \infty$. This is trivial for $\tau$, and we note $\tau > 0$. Consider $\rho$. We have for any $\boldsymbol{r} \in \mathbb{R}^m$,

$$\frac{1}{\sqrt{m}}\|\varphi(g(\boldsymbol{r}+\boldsymbol{b}))\| \leq \frac{1}{\sqrt{m}}\|\varphi(g(\boldsymbol{r}+\boldsymbol{b})) - \varphi(g(\boldsymbol{0}))\| + \frac{1}{\sqrt{m}}\|\varphi(g(\boldsymbol{0}))\|$$

$$\leq \frac{C}{\sqrt{m}}\|\boldsymbol{r}+\boldsymbol{b}\| + \frac{1}{\sqrt{m}}\|\varphi(g(\boldsymbol{0}))\|$$

$$\leq \frac{C}{\sqrt{m}}\|\boldsymbol{r}\| + \frac{C}{\sqrt{m}}\|\boldsymbol{b}\| + \frac{1}{\sqrt{m}}\|\varphi(g(\boldsymbol{0}))\|$$

$$\leq \frac{C}{\sqrt{m}}\|\boldsymbol{r}\| + C \quad (7)$$

for sufficiently large $m$. It is then easy to see that $\rho$ is uniformly bounded. Regarding $\gamma$,

$$|\gamma|\tau \leq \mathbb{E}_{\tilde{\boldsymbol{z}}}\left\{\frac{1}{m}\|\tilde{\boldsymbol{z}}\|\|\varphi(g(\tau\tilde{\boldsymbol{z}}+\boldsymbol{b}))\|\right\} \leq \sqrt{\mathbb{E}_{\tilde{\boldsymbol{z}}}\left\{\frac{1}{m}\|\tilde{\boldsymbol{z}}\|^2\right\}}\rho = \sqrt{\rho},$$

by Cauchy-Schwarz inequality. Therefore $\gamma$ is also uniformly bounded.

**Step 3.** We analyze the first term in $\Phi_n$. Notice that $\boldsymbol{y} \stackrel{\mathrm{d}}{=} \tau\tilde{\boldsymbol{z}}$ for some $\tilde{\boldsymbol{z}} \sim \mathcal{N}(0, I_m)$. In addition, the mapping $\boldsymbol{y} \mapsto \langle \boldsymbol{y}, \varphi(g(\boldsymbol{y}+\boldsymbol{b}))\rangle/m$ is uniformly pseudo-Lipschitz, by noticing that

$$\left|\frac{1}{m}\langle \boldsymbol{y}_1, \varphi(g(\boldsymbol{y}_1+\boldsymbol{b}))\rangle - \frac{1}{m}\langle \boldsymbol{y}_2, \varphi(g(\boldsymbol{y}_2+\boldsymbol{b}))\rangle\right|$$

$$\leq \frac{1}{m}\|\varphi(g(\boldsymbol{y}_1+\boldsymbol{b}))\|\|\boldsymbol{y}_1-\boldsymbol{y}_2\| + \frac{1}{m}\|\boldsymbol{y}_2\|\|\varphi(g(\boldsymbol{y}_1+\boldsymbol{b})) - \varphi(g(\boldsymbol{y}_2+\boldsymbol{b}))\|,$$

along with the fact that $\boldsymbol{y} \mapsto \varphi(g(\boldsymbol{y}+\boldsymbol{b}))$ is uniformly Lipschitz, and Eq. (7). As such, by Theorem 7,

$$\mathrm{Var}\left\{\frac{1}{m}\langle \boldsymbol{y}, \varphi(g(\boldsymbol{y}+\boldsymbol{b}))\rangle\right\} \leq \frac{C\tau^2}{m}\mathbb{E}\left\{\left(1 + \frac{1}{\sqrt{m}}\|\boldsymbol{y}\| + \frac{1}{\sqrt{m}}\|\boldsymbol{b}\|\right)^2\right\}$$

$$\leq \frac{3C\tau^2}{m}\mathbb{E}\left\{1 + \frac{1}{m}\|\boldsymbol{y}\|^2 + \frac{1}{m}\|\boldsymbol{b}\|^2\right\} = O\left(\frac{1}{m}\right),$$

which tends to 0 as $n \to \infty$. By Chebyshev's inequality, we thus have

$$\frac{1}{m}\langle \boldsymbol{y}, \varphi(g(\boldsymbol{y}+\boldsymbol{b}))\rangle \simeq \mathbb{E}\left\{\frac{1}{m}\langle \boldsymbol{y}, \varphi(g(\boldsymbol{y}+\boldsymbol{b}))\rangle\right\} = \mathbb{E}\left\{\frac{1}{m}\langle \tau\tilde{\boldsymbol{z}}, \varphi(g(\tau\tilde{\boldsymbol{z}}+\boldsymbol{b}))\rangle\right\} = \gamma\tau^2.$$

**Step 4.** We analyze the second term in $\Phi_n$. We have:

$$P_{\sigma(\boldsymbol{u})}^{\perp} \tilde{\boldsymbol{W}}^{\top} \varphi\left(g\left(\boldsymbol{y}+\boldsymbol{b}\right)\right) = \tilde{\boldsymbol{W}}^{\top} \varphi\left(g\left(\boldsymbol{y}+\boldsymbol{b}\right)\right) - \frac{\sigma\left(\boldsymbol{u}\right)}{\left\|\sigma\left(\boldsymbol{u}\right)\right\|^{2}} \left\langle \sigma\left(\boldsymbol{u}\right), \tilde{\boldsymbol{W}}^{\top} \varphi\left(g\left(\boldsymbol{y}+\boldsymbol{b}\right)\right) \right\rangle. \quad (8)$$

Note that the mapping $\boldsymbol{y} \mapsto \frac{1}{m} \left\|\varphi\left(g\left(\boldsymbol{y}+\boldsymbol{b}\right)\right)\right\|^{2}$ is uniformly pseudo-Lipschitz by a similar argument. Hence again by Theorem 7, $\operatorname{Var}\left\{\frac{1}{m} \left\|\varphi\left(g\left(\boldsymbol{y}+\boldsymbol{b}\right)\right)\right\|^{2}\right\} = O\left(1/m\right)$, which yields

$$\frac{1}{m} \left\|\varphi\left(g\left(\boldsymbol{y}+\boldsymbol{b}\right)\right)\right\|^{2} \simeq \mathbb{E}\left\{\frac{1}{m} \left\|\varphi\left(g\left(\boldsymbol{y}+\boldsymbol{b}\right)\right)\right\|^{2}\right\} = \mathbb{E}\left\{\frac{1}{m} \left\|\varphi\left(g\left(\tau\tilde{\boldsymbol{z}}+\boldsymbol{b}\right)\right)\right\|^{2}\right\} = \rho.$$

Recall that $\tilde{\boldsymbol{W}}$ is independent of $\mathcal{F}$, and as such, there exists a random variable $z' \sim \mathcal{N}\left(0,1\right)$ independent of $\boldsymbol{y}$ such that

$$\frac{1}{n} \left\langle \sigma\left(\boldsymbol{u}\right), \tilde{\boldsymbol{W}}^{\top} \varphi\left(g\left(\boldsymbol{y}+\boldsymbol{b}\right)\right) \right\rangle = \frac{1}{n\sqrt{n}} \left\|\sigma\left(\boldsymbol{u}\right)\right\| \left\|\varphi\left(g\left(\boldsymbol{y}+\boldsymbol{b}\right)\right)\right\| \sigma_W z' \simeq \frac{1}{n} \left\|\sigma\left(\boldsymbol{u}\right)\right\| \sqrt{\alpha\rho} \sigma_W z',$$

which converges to 0 in probability, where we have used the fact that $\rho$ is uniformly bounded from Step 2. Furthermore, there also exists a random vector $\boldsymbol{z} \sim \mathcal{N}\left(0, \boldsymbol{I}_n\right)$ independent of $\mathcal{F}$ such that

$$\tilde{\boldsymbol{W}}^{\top} \varphi\left(g\left(\boldsymbol{y}+\boldsymbol{b}\right)\right) = \frac{1}{\sqrt{n}} \left\|\varphi\left(g\left(\boldsymbol{y}+\boldsymbol{b}\right)\right)\right\| \sigma_W \boldsymbol{z}. \quad (9)$$

**Step 5.** We finish the proof. From the definition of uniform pseudo-Lipschitz functions and Eq. (8) and (9), we obtain:

$$\left|\Phi_n - \phi_n\left(\alpha\sigma_W^2 \gamma\sigma\left(\boldsymbol{u}\right) + \sqrt{\alpha\sigma_W^2 \rho}\boldsymbol{z} + \boldsymbol{v}\right)\right|$$

$$\leq C\left[1 + \left(\frac{\left|\left\langle\boldsymbol{y}, \varphi\left(g\left(\boldsymbol{y}+\boldsymbol{b}\right)\right)\right\rangle\right|}{\left\|\sigma\left(\boldsymbol{u}\right)\right\|^2} + \alpha\sigma_W^2\left|\gamma\right|\right)\frac{\left\|\sigma\left(\boldsymbol{u}\right)\right\|}{\sqrt{n}} + \left\|P_{\sigma(\boldsymbol{u})}^{\perp}\right\|_2 \left\|\tilde{\boldsymbol{W}}\right\|_2 \frac{\left\|\varphi\left(g\left(\boldsymbol{y}+\boldsymbol{b}\right)\right)\right\|}{\sqrt{n}}\right.$$

$$\left.+ \sqrt{\alpha\sigma_W^2 \rho}\frac{\left\|\boldsymbol{z}\right\|}{\sqrt{n}} + 2\frac{\left\|\boldsymbol{v}\right\|}{\sqrt{n}}\right]$$

$$\times \left[\left|\frac{\left\langle\boldsymbol{y}, \varphi\left(g\left(\boldsymbol{y}+\boldsymbol{b}\right)\right)\right\rangle}{\left\|\sigma\left(\boldsymbol{u}\right)\right\|^2} - \alpha\sigma_W^2\gamma\right|\frac{\left\|\sigma\left(\boldsymbol{u}\right)\right\|}{\sqrt{n}} + \left|\frac{1}{\sqrt{n}}\left\|\varphi\left(g\left(\boldsymbol{y}+\boldsymbol{b}\right)\right)\right\| - \sqrt{\alpha\rho}\right|\sigma_W\frac{\left\|\boldsymbol{z}\right\|}{\sqrt{n}}\right.$$

$$\left.+ \frac{\sqrt{n}}{\left\|\sigma\left(\boldsymbol{u}\right)\right\|}\left|\frac{1}{n}\left\langle\sigma\left(\boldsymbol{u}\right), \tilde{\boldsymbol{W}}^{\top}\varphi\left(g\left(\boldsymbol{y}+\boldsymbol{b}\right)\right)\right\rangle\right|\right].$$

Here notice that $\left\|P_{\sigma(\boldsymbol{u})}^{\perp}\right\|_2 = 1$, and as a standard fact from the random matrix theory, $\left\|\tilde{\boldsymbol{W}}\right\|_2 \leq c\left(\alpha, \sigma_W^2\right)$ a constant with high probability (see e.g. Vershynin (2012)). Furthermore, $\frac{1}{\sqrt{n}}\left\|\boldsymbol{z}\right\| \leq C$ with high probability due to the law of large numbers. Combining with the facts from Step 2, 3 and 4, it is then easy to see that

$$\left|\Phi_n - \phi_n\left(\alpha\sigma_W^2 \gamma\sigma\left(\boldsymbol{u}\right) + \sqrt{\alpha\sigma_W^2 \rho}\boldsymbol{z} + \boldsymbol{v}\right)\right| \simeq 0. \quad (10)$$

Finally, since $\gamma$ and $\rho$ are uniformly bounded from Step 2, it is easy to show that the mapping

$$\boldsymbol{z} \mapsto \phi_n\left(\alpha\sigma_W^2 \gamma\sigma\left(\boldsymbol{u}\right) + \sqrt{\alpha\sigma_W^2 \rho}\boldsymbol{z} + \boldsymbol{v}\right)$$

is uniformly pseudo-Lipschitz. Hence by Theorem 7,

$$\operatorname{Var}\left\{\phi_n\left(\alpha\sigma_W^2 \gamma\sigma\left(\boldsymbol{u}\right) + \sqrt{\alpha\sigma_W^2 \rho}\boldsymbol{z} + \boldsymbol{v}\right)\right\} = O\left(\frac{1}{n}\right),$$

which yields

$$\phi_n\left(\alpha\sigma_W^2 \gamma\sigma\left(\boldsymbol{u}\right) + \sqrt{\alpha\sigma_W^2 \rho}\boldsymbol{z} + \boldsymbol{v}\right) \simeq \mathbb{E}\left\{\phi_n\left(\alpha\sigma_W^2 \gamma\sigma\left(\boldsymbol{u}\right) + \sqrt{\alpha\sigma_W^2 \rho}\boldsymbol{z} + \boldsymbol{v}\right)\right\}.$$

Together with Eq. (6) and (10), this completes the proof.

### A.4 PROOF OF COROLLARY 6

By Proposition 5, for any sequence of uniformly pseudo-Lipschitz $\phi_n : \mathbb{R}^n \mapsto \mathbb{R}$:

$$\phi_n\left(f\left(\boldsymbol{u}\right)\right) \simeq \mathbb{E}_{\boldsymbol{z}}\left\{\phi_n\left(\alpha\sigma_W^2 \underline{\gamma}\sigma\left(\boldsymbol{u}\right) + \sqrt{\alpha\sigma_W^2\underline{\rho}}\boldsymbol{z} + \boldsymbol{v}\right)\right\},$$

for $\boldsymbol{z} \sim \mathcal{N}\left(0, \boldsymbol{I}_n\right)$ independent of $\boldsymbol{u}$, $\boldsymbol{v}$ and $\boldsymbol{b}$, in which

$$\underline{\gamma} = \underline{\gamma}\left(\underline{\tau}, \boldsymbol{b}\right) = \mathbb{E}_{\tilde{\boldsymbol{z}}}\left\{\frac{1}{m\underline{\tau}^2}\left\langle\underline{\tau}\tilde{\boldsymbol{z}}, \varphi\left(g\left(\underline{\tau}\tilde{\boldsymbol{z}} + \boldsymbol{b}\right)\right)\right\rangle\right\},$$

$$\underline{\rho} = \underline{\rho}\left(\underline{\tau}, \boldsymbol{b}\right) = \mathbb{E}_{\tilde{\boldsymbol{z}}}\left\{\frac{1}{m}\left\|\varphi\left(g\left(\underline{\tau}\tilde{\boldsymbol{z}} + \boldsymbol{b}\right)\right)\right\|^2\right\},$$

$$\underline{\tau}^2 = \underline{\tau}^2\left(\boldsymbol{u}\right) = \sigma_W^2\frac{\left\|\sigma\left(\boldsymbol{u}\right)\right\|^2}{n},$$

for $\tilde{\boldsymbol{z}} \sim \mathcal{N}\left(0, \boldsymbol{I}_m\right)$ independent of $\boldsymbol{u}$ and $\boldsymbol{b}$. It is easy to see that $\underline{\tau}^2 \simeq \tau^2 = \bar{\tau}^2 - \sigma_b^2$. For any $t \in (0, \infty)$, the mapping $\boldsymbol{b} \mapsto \underline{\gamma}(t, \boldsymbol{b})$ is uniformly pseudo-Lipschitz, and hence by Theorem 7, $\mathrm{Var}_{\boldsymbol{b}}\left\{\underline{\gamma}(t, \boldsymbol{b})\right\} = O\left(1/m\right)$, which yields $\underline{\gamma}(t, \boldsymbol{b}) \simeq \mathbb{E}_{\boldsymbol{b}}\left\{\underline{\gamma}(t, \boldsymbol{b})\right\}$. Recall $\underline{\tau}$ is independent of $\boldsymbol{b}$. As such, $\underline{\gamma}(\underline{\tau}, \boldsymbol{b}) \simeq \mathbb{E}_{\boldsymbol{b}}\left\{\underline{\gamma}(\underline{\tau}, \boldsymbol{b})\right\}$. Furthermore, the mapping $t \mapsto \mathbb{E}_{\boldsymbol{b}}\left\{\underline{\gamma}(t, \boldsymbol{b})\right\}$ is continuous and hence $\mathbb{E}_{\boldsymbol{b}}\left\{\underline{\gamma}(\underline{\tau}, \boldsymbol{b})\right\} \simeq \mathbb{E}_{\boldsymbol{b}}\left\{\underline{\gamma}(\tau, \boldsymbol{b})\right\}$. Performing a similar argument for $\underline{\rho}(\underline{\tau}, \boldsymbol{b})$, we thus have:

$$\underline{\rho} \simeq \rho,$$

$$\underline{\gamma} \simeq \mathbb{E}_{\tilde{\boldsymbol{z}}, \boldsymbol{b}}\left\{\frac{1}{m\tau^2}\left\langle\tau\tilde{\boldsymbol{z}}, \varphi\left(g\left(\tau\tilde{\boldsymbol{z}} + \boldsymbol{b}\right)\right)\right\rangle\right\}.$$

We note by Stein's lemma,

$$\mathbb{E}_{\tilde{\boldsymbol{z}}, \boldsymbol{b}}\left\{\left\langle\tau\tilde{\boldsymbol{z}}, \varphi\left(g\left(\tau\tilde{\boldsymbol{z}} + \boldsymbol{b}\right)\right)\right\rangle\right\} = \tau^2\mathbb{E}_{\tilde{\boldsymbol{z}}}\left\{\mathrm{div}\left(\varphi \circ g\right)\left(\sqrt{\tau^2 + \sigma_b^2}\tilde{\boldsymbol{z}}\right)\right\},$$

$$\mathbb{E}_{\tilde{\boldsymbol{z}}, \boldsymbol{b}}\left\{\left\langle\boldsymbol{b}, \varphi\left(g\left(\tau\tilde{\boldsymbol{z}} + \boldsymbol{b}\right)\right)\right\rangle\right\} = \sigma_b^2\mathbb{E}_{\tilde{\boldsymbol{z}}}\left\{\mathrm{div}\left(\varphi \circ g\right)\left(\sqrt{\tau^2 + \sigma_b^2}\tilde{\boldsymbol{z}}\right)\right\}.$$

Also notice that

$$\mathbb{E}_{\tilde{\boldsymbol{z}}, \boldsymbol{b}}\left\{\left\langle\tau\tilde{\boldsymbol{z}} + \boldsymbol{b}, \varphi\left(g\left(\tau\tilde{\boldsymbol{z}} + \boldsymbol{b}\right)\right)\right\rangle\right\} = \mathbb{E}_{\tilde{\boldsymbol{z}}}\left\{\left\langle\sqrt{\tau^2 + \sigma_b^2}\tilde{\boldsymbol{z}}, \varphi\left(g\left(\sqrt{\tau^2 + \sigma_b^2}\tilde{\boldsymbol{z}}\right)\right)\right\rangle\right\}.$$

Direct algebras then yield

$$\mathbb{E}_{\tilde{\boldsymbol{z}}, \boldsymbol{b}}\left\{\frac{1}{m\tau^2}\left\langle\tau\tilde{\boldsymbol{z}}, \varphi\left(g\left(\tau\tilde{\boldsymbol{z}} + \boldsymbol{b}\right)\right)\right\rangle\right\} = \gamma,$$

and hence $\underline{\gamma} \simeq \gamma$. Therefore,

$$\phi_n\left(f\left(\boldsymbol{u}\right)\right) \simeq \mathbb{E}_{\boldsymbol{z}}\left\{\phi_n\left(\alpha\sigma_W^2\gamma\sigma\left(\boldsymbol{u}\right) + \sqrt{\alpha\sigma_W^2\rho}\boldsymbol{z} + \boldsymbol{v}\right)\right\} \equiv h\left(\boldsymbol{u}, \boldsymbol{v}\right).$$

The mapping $\left(\boldsymbol{u}, \boldsymbol{v}\right) \mapsto h\left(\boldsymbol{u}, \boldsymbol{v}\right)$ is uniformly pseudo-Lipschitz and hence by Theorem 7, $\mathrm{Var}_{\boldsymbol{u}, \boldsymbol{v}}\left\{h\left(\boldsymbol{u}, \boldsymbol{v}\right)\right\} = O\left(1/n\right)$, which implies

$$h\left(\boldsymbol{u}, \boldsymbol{v}\right) \simeq \mathbb{E}_{\boldsymbol{u}, \boldsymbol{v}, \boldsymbol{z}}\left\{\phi_n\left(\alpha\sigma_W^2\gamma\sigma\left(\boldsymbol{u}\right) + \sqrt{\alpha\sigma_W^2\rho}\boldsymbol{z} + \boldsymbol{v}\right)\right\}$$

$$= \mathbb{E}_{\boldsymbol{z}, \boldsymbol{z}'}\left\{\phi_n\left(\alpha\sigma_W^2\gamma\sigma\left(\sigma_u\boldsymbol{z}\right) + \sqrt{\alpha\sigma_W^2\rho + \sigma_v^2}\boldsymbol{z}'\right)\right\}$$

which follows from the fact that $\boldsymbol{u}$, $\boldsymbol{v}$ and $\boldsymbol{z}$ are independent and normally distributed. The first claim is hence proven.

Eq. (4) follows immediately from the above. The proof of Eq. (5) is similar and hence omitted.

A.5 PROOF OF THEOREM 1

The following lemma states that, roughly speaking, $g_\ell$ is uniformly Lipschitz (indexed by $n$) with high probability. First, recall that $\boldsymbol{W}_\ell = \boldsymbol{W}_\ell(n)$ is indexed by $n$, and one can easily construct a joint probability space on which the sequence $\{\boldsymbol{W}_\ell(n)\}_{n \geq 1}$ is defined. The exact construction is immaterial, so long as for each $n$, the marginal distribution satisfies the setting in Section 2.1. We work in this joint space in the following.

**Lemma 8.** *For each $\ell = 1, ..., L$, there exists a finite constant $c_\ell > 0$ such that $\mathbb{P}\{\mathcal{E}_\ell^N\} \to 0$ as $N \to \infty$, where the event $\mathcal{E}_\ell^N$ is defined as*

$$\mathcal{E}_\ell^N = \left\{ \sup_{(n,\boldsymbol{u}_1,\boldsymbol{u}_2) \in \mathcal{D}_\ell} \frac{\|g_\ell(\boldsymbol{u}_1) - g_\ell(\boldsymbol{u}_2)\|}{\|\boldsymbol{u}_1 - \boldsymbol{u}_2\|} > c_\ell \right\}, \quad \mathcal{D}_\ell = \{n > N, \, \boldsymbol{u}_1, \boldsymbol{u}_2 \in \mathbb{R}^{n_{\ell-1}}, \, \boldsymbol{u}_1 \neq \boldsymbol{u}_2\}.$$

*Proof.* Consider $\ell = L$. Recall that $g_L(\boldsymbol{u}) = \boldsymbol{W}_L^\top \varphi_L(\boldsymbol{W}_L \sigma_{L-1}(\boldsymbol{u}) + \boldsymbol{b}_L) + \boldsymbol{v}_L$. Then for any $\boldsymbol{u}_1$ and $\boldsymbol{u}_2$,

$$\begin{aligned}
\|g_L(\boldsymbol{u}_1) - g_L(\boldsymbol{u}_2)\| &= \left\| \boldsymbol{W}_L^\top \varphi_L(\boldsymbol{W}_L \sigma_{L-1}(\boldsymbol{u}_1) + \boldsymbol{b}_L) - \boldsymbol{W}_L^\top \varphi_L(\boldsymbol{W}_L \sigma_{L-1}(\boldsymbol{u}_2) + \boldsymbol{b}_L) \right\| \\
&\leq \|\boldsymbol{W}_L\|_2 \|\varphi_L(\boldsymbol{W}_L \sigma_{L-1}(\boldsymbol{u}_1) + \boldsymbol{b}_L) - \varphi_L(\boldsymbol{W}_L \sigma_{L-1}(\boldsymbol{u}_2) + \boldsymbol{b}_L)\| \\
&\overset{(i)}{\leq} C \|\boldsymbol{W}_L\|_2 \|\boldsymbol{W}_L \sigma_{L-1}(\boldsymbol{u}_1) - \boldsymbol{W}_L \sigma_{L-1}(\boldsymbol{u}_2)\| \\
&\leq C \|\boldsymbol{W}_L\|_2^2 \|\sigma_{L-1}(\boldsymbol{u}_1) - \sigma_{L-1}(\boldsymbol{u}_2)\| \\
&\overset{(ii)}{\leq} C \|\boldsymbol{W}_L\|_2^2 \|\boldsymbol{u}_1 - \boldsymbol{u}_2\|,
\end{aligned}$$

where steps $(i)$ and $(ii)$ are because $\varphi_L$ and $\sigma_{L-1}$ are Lipschitz. It is a standard fact in the random matrix theory that, for each $\ell = 1, ..., L$, there exist finite constants $\eta_\ell = \eta_\ell\left(\alpha_\ell, \sigma_{W,\ell}^2\right) > 0$ and $c_\ell' = c_\ell'\left(\alpha_\ell, \sigma_{W,\ell}^2\right) > 0$ such that $\mathbb{P}\left\{\|\boldsymbol{W}_\ell\|_2^2 > \eta_\ell\right\} \leq e^{-c_\ell' n}$ (see e.g. Vershynin (2012)). As such, using this fact for $\ell = L$, by the union bound,

$$\mathbb{P}\{\mathcal{E}_L^N\} \leq \sum_{n > N} e^{-c_L' n} \leq \int_{u=N}^{\infty} e^{-c_L' u} du = \frac{1}{c_L'} e^{-c_L' N},$$

which proves the claim for $\ell = L$. To see the claim for general $\ell$, we have from a similar calculation:

$$\|g_\ell(\boldsymbol{u}_1) - g_\ell(\boldsymbol{u}_2)\| \leq \|\boldsymbol{W}_\ell\|_2 \|g_{\ell+1}(\boldsymbol{W}_\ell \sigma_{\ell-1}(\boldsymbol{u}_1) + \boldsymbol{b}_\ell) - g_{\ell+1}(\boldsymbol{W}_\ell \sigma_{\ell-1}(\boldsymbol{u}_2) + \boldsymbol{b}_\ell)\|$$

$$\leq C^{L-\ell+1} \left( \prod_{r=\ell}^{L} \|\boldsymbol{W}_r\|_2^2 \right) \|\boldsymbol{u}_1 - \boldsymbol{u}_2\|.$$

Therefore,

$$\sup_{(n,\boldsymbol{u}_1,\boldsymbol{u}_2) \in \mathcal{D}_\ell} \frac{\|g_\ell(\boldsymbol{u}_1) - g_\ell(\boldsymbol{u}_2)\|}{\|\boldsymbol{u}_1 - \boldsymbol{u}_2\|} \leq C^{L-\ell+1} \left( \prod_{r=\ell}^{L} \eta_r \right)$$

with probability at least

$$1 - \sum_{n > N} \sum_{r=\ell}^{L} e^{-c_r' N} \geq 1 - \frac{L}{c'} e^{-c' N}, \qquad c' = \min_{1 \leq \ell \leq L} c_\ell',$$

by the union bound. This proves the claim. $\square$

**Lemma 9.** *For each $\ell \geq 0$, $\tau_{\ell+1}$ and $\bar{\tau}_{\ell+1}$ are finite and strictly positive. Furthermore, $\frac{1}{n_\ell} \|\sigma_\ell(\boldsymbol{x}_\ell)\|^2 \simeq \tau_{\ell+1}^2 / \sigma_{W,\ell+1}^2$, and $\boldsymbol{x}_{\ell+1} \cong \bar{\tau}_{\ell+1} \boldsymbol{z}$ for $\boldsymbol{z} \sim \mathcal{N}\left(0, \boldsymbol{I}_{n_{\ell+1}}\right)$.*

*Proof.* We prove the lemma by induction. The claim is trivially true for $\ell = 0$ by assumption. Assume the claim for some $\ell \geq 0$. Due to independence, conditioning on $\boldsymbol{x}_\ell$,

$$\boldsymbol{x}_{\ell+1}|\boldsymbol{x}_\ell \overset{\mathrm{d}}{=} \sqrt{\frac{\sigma_{W,\ell+1}^2}{n_\ell} \|\sigma_\ell(\boldsymbol{x}_\ell)\|^2 + \sigma_{b,\ell+1}^2} \boldsymbol{z},$$

where $\boldsymbol{z} \sim \mathcal{N}\left(0, \boldsymbol{I}_{n_{\ell+1}}\right)$ independent of $\boldsymbol{x}_\ell$. Since $\frac{1}{n_\ell}\left\|\sigma_\ell\left(\boldsymbol{x}_\ell\right)\right\|^2 \simeq \tau_{\ell+1}^2 / \sigma_{W,\ell+1}^2$, we then have

$$\boldsymbol{x}_{\ell+1} \cong \sqrt{\tau_{\ell+1}^2 + \sigma_{b,\ell+1}^2}\, \boldsymbol{z} = \bar{\tau}_{\ell+1} \boldsymbol{z},$$

which implies

$$\frac{1}{n_{\ell+1}}\left\|\sigma_{\ell+1}\left(\boldsymbol{x}_{\ell+1}\right)\right\|^2 \to \mathbb{E}_{z'}\left\{\sigma_{\ell+1}\left(\bar{\tau}_{\ell+1} z'\right)^2\right\}$$

for $z' \sim \mathcal{N}(0,1)$. By the induction hypothesis, $\bar{\tau}_{\ell+1} > 0$. Hence by assumption, the right-hand side is finite and strictly positive. One also recognizes that it is equal to $\tau_{\ell+2}^2 / \sigma_{W,\ell+2}^2$. Since $\sigma_{W,\ell+2}$ is strictly positive and finite, so is $\tau_{\ell+2}$, and hence so is $\bar{\tau}_{\ell+2}$. This completes the proof. $\qquad\square$

We are now ready for the proof of Theorem 1. Claim (a) of the theorem, which follows directly from Lemma 9, is just a forward pass through the encoder and hence is the same as in the case of random feedforward networks (without weight tying) Poole et al. (2016).

For $\ell = 2, ..., L$, let $\mathcal{H}_\ell$ denote the following three claims:

$$(1): \quad \hat{\boldsymbol{x}}_\ell \cong \alpha_\ell \sigma_{W,\ell}^2 \gamma_{\ell+1} \sigma_{\ell-1}\left(\bar{\tau}_{\ell-1} \boldsymbol{z}_1\right) + \sqrt{\alpha_\ell \sigma_{W,\ell}^2 \rho_{\ell+1} + \sigma_{v,\ell}^2}\, \boldsymbol{z}_2,$$

$$(2): \quad \frac{1}{n_{\ell-1}}\left\|\varphi_{\ell-1}\left(g_\ell\left(\bar{\tau}_{\ell-1} \tilde{\boldsymbol{z}}_{\ell-1}\right)\right)\right\|^2 \simeq \rho_\ell,$$

$$(3): \quad \frac{1}{n_{\ell-1} \bar{\tau}_{\ell-1}^2}\left\langle \bar{\tau}_{\ell-1} \tilde{\boldsymbol{z}}_{\ell-1}, \varphi_{\ell-1}\left(g_\ell\left(\bar{\tau}_{\ell-1} \tilde{\boldsymbol{z}}_{\ell-1}\right)\right)\right\rangle \simeq \gamma_\ell,$$

for $\boldsymbol{z}_1, \boldsymbol{z}_2, \sim \mathcal{N}\left(0, \boldsymbol{I}_{n_{\ell-1}}\right)$ independently, and $\tilde{\boldsymbol{z}}_{\ell-1} \sim \mathcal{N}\left(0, \boldsymbol{I}_{n_{\ell-1}}\right)$ independent of $g_\ell$, with the understanding that $g_{L+1}$ is the identity. We prove them by induction, and Claim (b) then follows immediately. We first note that with high probability, for any $\ell$, $\frac{1}{n_\ell}\left\|\boldsymbol{b}_\ell\right\|^2$ and $\frac{1}{n_{\ell-1}}\left\|\boldsymbol{v}_\ell\right\|^2$ are bounded by the law of large numbers. Consider $\mathcal{H}_L$. Note that $\boldsymbol{x}_{L-1}$ is independent of $g_L$, and that $\frac{1}{n_{L-1}}\left\|\sigma_{L-1}\left(\boldsymbol{x}_{L-1}\right)\right\|^2$ converges to a finite non-zero constant in probability by Lemma 9. Hence by Corollary 6, with $g$ being the identity, $f$ being $g_L$, $\boldsymbol{u}$ being $\boldsymbol{x}_{L-1}$ (recalling $\hat{\boldsymbol{x}}_L = g_L\left(\boldsymbol{x}_{L-1}\right)$) and $\phi$ being $\varphi_{L-1}$, and the fact $\boldsymbol{x}_{L-1} \cong \bar{\tau}_{L-1} \tilde{\boldsymbol{z}}_{\ell-1}$ from Claim (a), $\mathcal{H}_L$ is proven.

Assuming $\mathcal{H}_{\ell+1}$ for some $\ell$, we prove $\mathcal{H}_\ell$. Recall that $g_{\ell+1}$ is independent of $\boldsymbol{\Theta}_\ell = \{\boldsymbol{W}_\ell, \boldsymbol{b}_\ell, \boldsymbol{v}_\ell, \boldsymbol{x}_{\ell-1}, \tilde{\boldsymbol{z}}_{\ell-1}\}$, and $g_\ell$ is independent of $\boldsymbol{x}_{\ell-1}$ and $\tilde{\boldsymbol{z}}_{\ell-1}$. Also from Claim (a), $\boldsymbol{x}_{\ell-1} \cong \bar{\tau}_{\ell-1} \tilde{\boldsymbol{z}}_{\ell-1}$. Consider some $N \in \mathbb{N}$ and let $\mathcal{E}_{\ell+1}^N$ denote the event as defined in Lemma 8. On the event $\neg\mathcal{E}_{\ell+1}^N$ (the complement of $\mathcal{E}_{\ell+1}^N$), by Corollary 6, with respect to the randomness of $\boldsymbol{\Theta}_\ell$,

$$\hat{\boldsymbol{x}}_\ell \cong \alpha_\ell \sigma_{W,\ell}^2 \gamma \sigma_{\ell-1}\left(\bar{\tau}_{\ell-1} \boldsymbol{z}_1\right) + \sqrt{\alpha_\ell \sigma_{W,\ell}^2 \rho + \sigma_{v,\ell}^2}\, \boldsymbol{z}_2,$$

$$\frac{1}{n_{\ell-1}}\left\|\varphi_{\ell-1}\left(g_\ell\left(\bar{\tau}_{\ell-1} \tilde{\boldsymbol{z}}_{\ell-1}\right)\right)\right\|^2 \simeq \mathbb{E}_{z_1, z_2}\left\{\varphi_{\ell-1}\left(\alpha_\ell \sigma_{W,\ell}^2 \gamma \sigma_{\ell-1}\left(\bar{\tau}_{\ell-1} \boldsymbol{z}_1\right) + \sqrt{\alpha_\ell \sigma_{W,\ell}^2 \rho + \sigma_{v,\ell}^2}\, \boldsymbol{z}_2\right)^2\right\},$$

$$\frac{1}{n_{\ell-1}}\left\langle \tilde{\boldsymbol{z}}_{\ell-1}, \varphi_{\ell-1}\left(g_\ell\left(\bar{\tau}_{\ell-1} \tilde{\boldsymbol{z}}_{\ell-1}\right)\right)\right\rangle \simeq \mathbb{E}_{z_1, z_2}\left\{z_1 \phi\left(\alpha_\ell \sigma_{W,\ell}^2 \gamma \sigma_{\ell-1}\left(\bar{\tau}_{\ell-1} \boldsymbol{z}_1\right) + \sqrt{\alpha_\ell \sigma_{W,\ell}^2 \rho + \sigma_{v,\ell}^2}\, \boldsymbol{z}_2\right)\right\},$$

in which

$$\gamma = \mathbb{E}_{\boldsymbol{z}'}\left\{\frac{1}{n_\ell \bar{\tau}_\ell^2}\left\langle \bar{\tau}_\ell \boldsymbol{z}', \varphi_\ell\left(g_{\ell+1}\left(\bar{\tau}_\ell \boldsymbol{z}'\right)\right)\right\rangle\right\},$$

$$\rho = \mathbb{E}_{\boldsymbol{z}'}\left\{\frac{1}{n_\ell}\left\|\varphi_\ell\left(g_{\ell+1}\left(\bar{\tau}_\ell \boldsymbol{z}'\right)\right)\right\|^2\right\}.$$

Note that here $\gamma$ and $\rho$ are functions of $g_{\ell+1}$ and hence random, and $\boldsymbol{z}' \sim \mathcal{N}\left(0, \boldsymbol{I}_{n_\ell}\right)$ independent of $g_{\ell+1}$. On the event $\neg\mathcal{E}_{\ell+1}^N$, with the fact that $\bar{\tau}_\ell$ is finite and non-zero by Lemma 9, we have the mapping

$$\boldsymbol{z}' \mapsto \frac{1}{n_\ell \bar{\tau}_\ell^2}\left\langle \bar{\tau}_\ell \boldsymbol{z}', \varphi_\ell\left(g_{\ell+1}\left(\bar{\tau}_\ell \boldsymbol{z}'\right)\right)\right\rangle \equiv h\left(\boldsymbol{z}'\right)$$

is uniformly pseudo-Lipschitz, and hence applying Theorem 7, one obtains that

$$\mathrm{Var}_{\boldsymbol{z}'}\left\{h\left(\boldsymbol{z}'\right)\right\} = O\left(\frac{1}{n_\ell}\right),$$

with the right-hand side independent of $g_{\ell+1}$ for $n > N$. Consequently, due to Lemma 8 and Chebyshev's inequality, for any $\epsilon > 0$,

$$
\mathbb{P}\left\{|h\left(\boldsymbol{z}'\right) - \gamma| > \epsilon, \ \neg\mathcal{E}_{\ell+1}^N\right\} = \mathbb{E}\left\{\mathbb{P}_{\boldsymbol{z}'}\left\{|h\left(\boldsymbol{z}'\right) - \gamma| > \epsilon\right\}\mathbb{I}_{\neg\mathcal{E}_{\ell+1}^N}\right\}
$$
$$
\leq \mathbb{E}\left\{\frac{1}{\epsilon^2}\operatorname{Var}_{\boldsymbol{z}'}\left\{h\left(\boldsymbol{z}'\right)\right\}\mathbb{I}_{\neg\mathcal{E}_{\ell+1}^N}\right\} = o_n\left(1\right)\left(1 - o_N\left(1\right)\right) = o_n\left(1\right),
$$

where $o_n\left(1\right) \to 0$ as $n \to \infty$. On the other hand, since $\boldsymbol{z}'$ is independent of $g_{\ell+1}$, invoking $\mathcal{H}_{\ell+1}$,

$$
\mathbb{P}\left\{|h\left(\boldsymbol{z}'\right) - \gamma_{\ell+1}| > \epsilon\right\} = o_n\left(1\right).
$$

As such,

$$
\mathbb{P}\left\{|\gamma - \gamma_{\ell+1}| > \epsilon\right\} \leq \mathbb{P}\left\{|h\left(\boldsymbol{z}'\right) - \gamma_{\ell+1}| > \epsilon\right\} + \mathbb{P}\left\{|h\left(\boldsymbol{z}'\right) - \gamma| > \epsilon\right\}
$$
$$
\leq \mathbb{P}\left\{|h\left(\boldsymbol{z}'\right) - \gamma_{\ell+1}| > \epsilon\right\} + \mathbb{P}\left\{|h\left(\boldsymbol{z}'\right) - \gamma| > \epsilon, \ \neg\mathcal{E}_{\ell+1}^N\right\} + \mathbb{P}\left\{\mathcal{E}_{\ell+1}^N\right\}
$$
$$
= o_n\left(1\right) + o_n\left(1\right) + o_N\left(1\right).
$$

Letting $n \to \infty$ then $N \to \infty$, we then have $\gamma \simeq \gamma_{\ell+1}$. Similarly, we also have $\rho \simeq \rho_{\ell+1}$. It is then easy to deduce $\mathcal{H}_\ell$, recalling the definitions of $\gamma_\ell$ and $\rho_\ell$.

The claim that $\bar{\tau}_{\ell-1}\boldsymbol{z}_1$ can be replaced with $\boldsymbol{x}_{\ell-1}$ in the expression for $\hat{\boldsymbol{x}}_\ell$ in Claim (b) can be recognized easily by doing the same replacement in the proof of Corollary 6 and the above proof. The proof of Claim (c) is similar. We omit these repetitive steps.

## B  NUMERICAL VERIFICATION OF THEOREM 1

We perform simple simulations to verify Theorem 1 at finite dimensions. In particular, we simulate a random weight-tied autoencoder, as described in Section 2.1, with $L = 50$, $\sigma_{W,\ell}^2 = 2.312$, $\sigma_{b,\ell}^2 = 0.211$, $\sigma_{v,\ell}^2 = 0$, $\varphi_\ell = \sigma_\ell = \tanh$, identity $\varphi_0$ and $\sigma_0$, $\alpha_\ell = 1$, and consequently $n_\ell = n_0$, for an input $\boldsymbol{x} \in \mathbb{R}^{n_0}$ whose first half of the entries are $+1$ and the rest are $-1$. Then we compute the following:

$$
\hat{\gamma}_{\ell+1} = \frac{\langle\hat{\boldsymbol{x}}_\ell, \sigma_{\ell-1}\left(\boldsymbol{x}_{\ell-1}\right)\rangle}{\alpha_\ell\sigma_{W,\ell}^2\|\sigma_{\ell-1}\left(\boldsymbol{x}_{\ell-1}\right)\|^2},
$$
$$
\hat{\rho}_{\ell+1} = \frac{1}{\alpha_\ell\sigma_{W,\ell}^2}\left(\frac{1}{n_{\ell-1}}\|\hat{\boldsymbol{x}}_\ell\|^2 - \frac{\alpha_\ell^2\sigma_{W,\ell}^4\hat{\gamma}_{\ell+1}^2}{n_{\ell-1}}\|\sigma_{\ell-1}\left(\boldsymbol{x}_{\ell-1}\right)\|^2 - \sigma_{v,\ell}^2\right).
$$

We also compute $\{\gamma_\ell, \rho_\ell\}_{\ell=2,\ldots,L+1}$ as in Section 2.2, and $\gamma$ and $\rho$ as in Section 3.1. Theorem 1 predicts that $\hat{\gamma}_\ell \simeq \gamma_\ell$ and $\hat{\rho}_\ell \simeq \rho_\ell$. Section 3.1 also asserts that $\gamma_\ell \approx \gamma$ and $\rho_\ell \approx \rho$ for $\ell \ll L$. Fig. 6 shows the results for $n_0 = 500$ and $n_0 = 2000$. We observe a quantitative agreement already for $n_0 = 500$, and this improves with larger $n_0$.

Next we verify the normality of the variation component in $\hat{\boldsymbol{x}}$. We compute

$$
\hat{\boldsymbol{z}} = \frac{\hat{\boldsymbol{x}} - \alpha_1\sigma_{W,1}^2\hat{\gamma}_2\boldsymbol{x}}{\sqrt{\alpha_1\sigma_{W,1}^2\hat{\rho}_2 + \sigma_{v,1}^2}}.
$$

Its empirical distribution should be close to $\mathcal{N}\left(0, 1\right)$, in light of Theorem 1. We make this comparison in Fig. 7, and again observe good agreement already for $n_0 = 500$.

Finally we re-simulate the autoencoder with different distributions of the weights $\boldsymbol{W}_\ell$. In particular, we try with the Bernoulli distribution, the uniform distribution, and the Laplace distribution, with their means and variances adjusted to zero and $\sigma_{W,\ell}^2/n_{\ell-1}$ respectively. The results for $n_0 = 2000$ are plotted in Fig. 8 and 9. We observe good quantitative agreements between the simulations for these non-Gaussian distributions and the prediction as in Theorem 1, although the theorem is proven only for Gaussian weights.

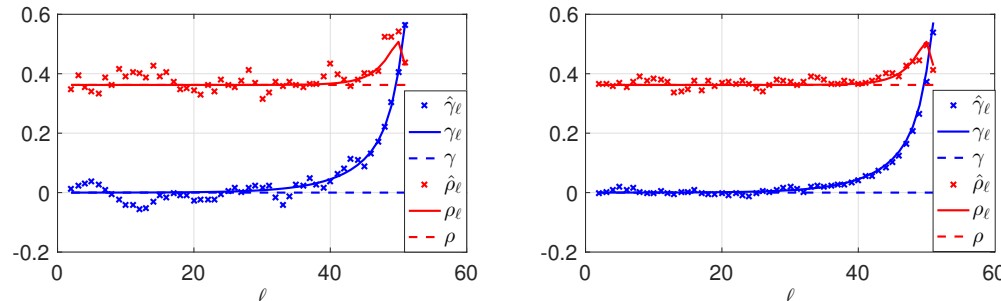

Figure 6: The agreement among $\hat{\gamma}_\ell$, $\gamma_\ell$ and $\gamma$, and among $\hat{\rho}_\ell$, $\rho_\ell$ and $\rho$, for $\ell = 2, ..., 51$ and Gaussian weights. The setting is described in Appendix B. We take a single run for the simulation of the autoencoder. Here $n_0 = 500$ (left) and $n_0 = 2000$ (right).

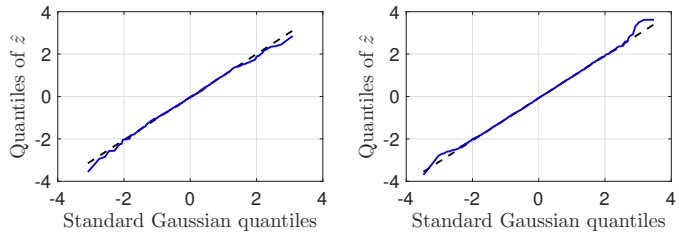

Figure 7: Quantile-quantile plots for the empirical distribution of $\hat{z}$, described in Appendix B, versus the standard Gaussian distribution. Here $n_0 = 500$ (left) and $n_0 = 2000$ (right).

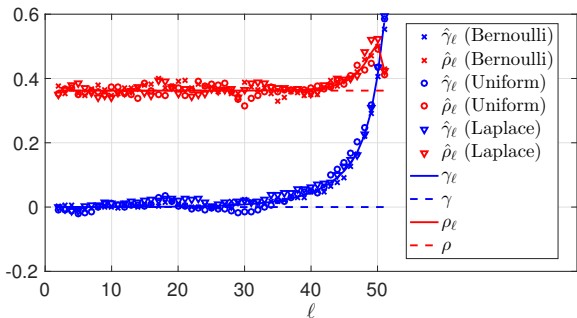

Figure 8: The agreement among $\hat{\gamma}_\ell$, $\gamma_\ell$ and $\gamma$, and among $\hat{\rho}_\ell$, $\rho_\ell$ and $\rho$, for $\ell = 2, ..., 51$, for different distributions of the weights. The setting is described in Appendix B. We take a single run for the simulation of the autoencoder. Here $n_0 = 2000$.

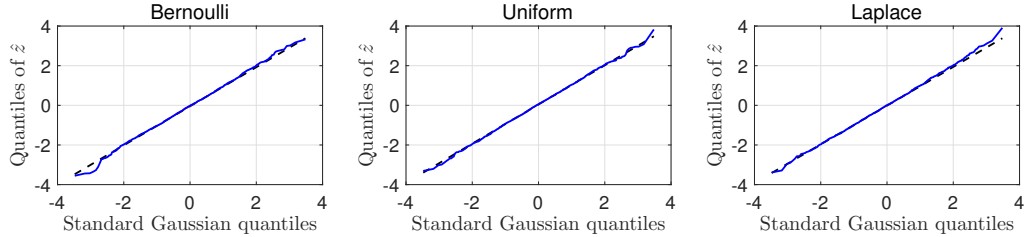

Figure 9: Quantile-quantile plots for the empirical distribution of $\hat{z}$, described in Appendix B, versus the standard Gaussian distribution, for different distributions of the weights. Here $n_0 = 2000$.

## C   ON THE FIXED POINT EQUATIONS OF $\gamma$ AND $\rho$

### C.1   COMPUTATION OF $\gamma$ AND $\rho$

Computing $\gamma$ and $\rho$ as from Eq. (2) and (3) amounts to evaluating double integrals. For simple $\varphi$, such as the ReLU, the integrals $f_\varphi(a, b) = \mathbb{E}\{\varphi(a + bz)\}$ and $g_\varphi(a, b) = \mathbb{E}\left\{\varphi(a + bz)^2\right\}$, for $z \sim \mathcal{N}(0, 1)$, can be calculated in closed forms. In such cases, one can make reduction to one-dimensional integrals:

$$\gamma = \frac{1}{\bar{\tau}^2}\mathbb{E}\left\{\bar{\tau}z_1 f_\varphi\left(\alpha\sigma_W^2\gamma\sigma(\bar{\tau}z_1), \sqrt{\alpha\sigma_W^2\rho}\right)\right\}, \quad \rho = \mathbb{E}\left\{g_\varphi\left(\alpha\sigma_W^2\gamma\sigma(\bar{\tau}z_1), \sqrt{\alpha\sigma_W^2\rho}\right)\right\}.$$

We proceed with computing $\gamma$ and $\rho$ as follows. From a random initialization $\gamma^{(0)} > 0$ and $\rho^{(0)} > 0$, one iteratively updates $\gamma^{(t+1)} = G\left(\gamma^{(t)}, \rho^{(t)}\right)$ and $\rho^{(t+1)} = R\left(\gamma^{(t)}, \rho^{(t)}\right)$, and stops when the incremental update is negligible or the number of iterations exceeds a threshold. Upon convergence, this procedure finds a stable fixed point.

### C.2   PROPERTIES OF $\gamma$ AND $\rho$

We prove several properties of $\gamma$ and $\rho$. We recall $G(\gamma, \rho)$ and $R(\gamma, \rho)$ from Eq. (2) and (3) for ease of reading:

$$G(\gamma, \rho) = \frac{1}{\bar{\tau}^2}\mathbb{E}\left\{\bar{\tau}z_1\varphi\left(\beta\gamma\sigma(\bar{\tau}z_1) + \sqrt{\beta\rho}z_2\right)\right\} \overset{(i)}{=} \beta\gamma\mathbb{E}\left\{\varphi'\left(\beta\gamma\sigma(\bar{\tau}z_1) + \sqrt{\beta\rho}z_2\right)\sigma'(\bar{\tau}z_1)\right\},$$

$$R(\gamma, \rho) = \mathbb{E}\left\{\varphi\left(\beta\gamma\sigma(\bar{\tau}z_1) + \sqrt{\beta\rho}z_2\right)^2\right\},$$

for $z_1, z_2 \sim \mathcal{N}(0, 1)$ independently, where $(i)$ is due to Stein's lemma. Recall that the fixed points equations are $\gamma = G(\gamma, \rho)$ and $\rho = R(\gamma, \rho)$. We also recall that $\beta = \alpha\sigma_W^2 > 0$. In light of Remark 3, we will consider Lipschitz continuous, non-decreasing $\varphi$ and $\sigma$, so that $\gamma \geq 0$ (and of course, $\rho \geq 0$). We will study these equations, first by stating some propositions, then discussing their implications, although we caution that the link between the propositions and the suggested implications is not entirely rigorous. All the proofs are deferred to Section C.3. We note that while the discussions concern with ReLU or $\tanh$ activations, the propositions apply to broader classes of functions.

In the following, when we say an initialization for $\gamma$ and $\rho$, we mean either an initialization in the context of an iterative process to find the fixed points as in Section C.1, or $\gamma_{L+1}$ and $\rho_{L+1}$ as in Section 2.2 in the context of autoencoders with $L \to \infty$. We also say $\gamma$ is a fixed point, without referencing to $\rho$, to mean that it is only a fixed point of $\gamma = G(\gamma, \rho)$ for a given $\rho$, and similarly for $\rho$. When we mention both $\gamma$ and $\rho$ as a fixed point, we mean a fixed point to both $\gamma = G(\gamma, \rho)$ and $\rho = R(\gamma, \rho)$. We will use $\partial_u^k f$ to denote the $k$th-order partial derivative of $f$ with respect to $u$.

#### C.2.1   THE CASE OF ReLU $\varphi$

The following result is exclusive to ReLU $\varphi$.

**Proposition 10.** *Consider that $\varphi$ is the ReLU and $\sigma$ is Lipschitz continuous and non-decreasing:*

(a) *$\gamma = 0$ and $\rho = 0$ is a fixed point. Furthermore, at $\gamma = 0$ and $\beta \neq 2$, the mapping $\rho \mapsto R(0, \rho)$ admits $\rho = 0$ as the only fixed point, which is stable if $\beta < 2$ and unstable if $\beta > 2$. Also at $\gamma = 0$ and $\beta = 2$, any $\rho$ is a stable fixed point.*

(b) *Assume $\sigma$ is positive on a set of positive Lebesgue measure. If $\gamma \to +\infty$, it must be that $\rho \to +\infty$.*

(c) *Consider $\bar{\tau} \in (0, \infty)$. Assume $\sigma$ is non-zero on a set of positive Lebesgue measure, and that $\sigma(u) = 0$ for all $u \leq 0$. Then no $\gamma > 0$ is a stable fixed point. Furthermore,*

- *if $\beta\mathbb{E}\{\sigma'(\bar{\tau}z_1)\} \leq 1$, there is only one fixed point at $\gamma = 0$, which is stable;*
- *if $\beta\mathbb{E}\{\sigma'(\bar{\tau}z_1)\} \in (1, 2)$, there are two: one at $\gamma = 0$, which is stable, and the other at $\gamma > 0$, which is unstable;*

- *if $\beta \mathbb{E} \left\{ \sigma' \left( \bar{\tau} z_1 \right) \right\} \geq 2$, there is only one fixed point at $\gamma = 0$, which is unstable.*

(d) *Assume $\sigma$ is non-zero on a set of positive Lebesgue measure, and that $\sigma$ is an odd function. Then for any $\rho$, the mapping $\gamma \mapsto G \left( \gamma, \rho \right)$ is a straight line through the point $(0, 0)$. Furthermore, if $\beta \mathbb{E} \left\{ \sigma' \left( \bar{\tau} z_1 \right) \right\} \neq 2$, there is only one fixed point at $\gamma = 0$, which is stable for $\beta \mathbb{E} \left\{ \sigma' \left( \bar{\tau} z_1 \right) \right\} < 2$ and unstable for $\beta \mathbb{E} \left\{ \sigma' \left( \bar{\tau} z_1 \right) \right\} > 2$; if $\beta \mathbb{E} \left\{ \sigma' \left( \bar{\tau} z_1 \right) \right\} = 2$, any $\gamma$ is a fixed point.*

(e) *Assume $\sigma$ is odd, and consider $\beta > 2$. Given $\gamma \geq 0$, we have $R \left( \gamma, \rho \right) > \rho$ for all $\rho \geq 0$.*

The proposition suggests the following picture. First consider $\sigma$ is ReLU. Since $\mathbb{E} \left\{ \sigma' \left( \bar{\tau} z_1 \right) \right\} = \mathbb{P} \left( \bar{\tau} z_1 \geq 0 \right) = 0.5$, we have two phase transitions at $\beta = 2$ and at $\beta = 4$. In particular, based on Proposition 10:

- When $\beta < 2$, with any initialization, we have convergence to $\gamma = 0$ and $\rho = 0$. This is based on Claim (a) and (c).

- When $\beta \in (2, 4)$, with certain initializations, we have convergence to $\gamma = 0$ and divergence to $\rho = +\infty$; with certain other initializations, we have divergence to $\gamma = +\infty$ and $\rho = +\infty$. This excludes a special initialization at the unstable fixed point, which is a singleton and essentially a rare case. This is based on Claim (a), (b) and (c).

- When $\beta \geq 4$, with any non-zero initialization, we have divergence to $\gamma = +\infty$ and hence $\rho = +\infty$. This is based on Claim (b) and (c).

Now we consider $\sigma$ is $\tanh$. Let $\beta_0 = \beta_0 \left( \bar{\tau} \right) = 2 / \mathbb{E} \left\{ \sigma' \left( \bar{\tau} z_1 \right) \right\}$, and it is easy to see that $\beta_0 > 2$ since $\sigma = \tanh$. The following picture is then expected:

- When $\beta < 2$, with any initialization, we have convergence to $\gamma = \rho = 0$. This is based on Claim (a) and (d).

- When $\beta \in (2, \beta_0)$, with any non-zero initialization, we have convergence to $\gamma = 0$ and divergence to $\rho = +\infty$. This is based on Claim (a) and (d).

- When $\beta > \beta_0$, with any non-zero initialization, we have divergence to $\gamma = +\infty$ and $\rho = +\infty$. This is based on Claim (b) and (d).

- When $\beta = \beta_0$, we have that $\gamma$ is unchanged from the initialization. Since $\beta_0 > 2$, we then have divergence to $\rho = +\infty$. This is based on Claim (b) and (e).

One crucial property of the ReLU is that it is unbounded at infinity and its derivative at infinity is bounded away from zero. This allows $\gamma$ and $\rho$ to grow to infinity. This is a stark contrast to the case $\varphi$ is bounded, for instance, $\varphi = \tanh$ as we shall see.

### C.2.2  THE CASE OF $\tanh$ $\varphi$

We state a result that is relevant to $\varphi = \tanh$.

**Proposition 11.** *Assume that $\varphi$ thrice-differentiable with $\varphi \left( 0 \right) = 0$, $\varphi' \left( 0 \right) = \kappa$, and $\left\| \varphi^{(k)} \right\|_\infty \leq C$ for $k = 0, ..., 3$, where $\varphi^{(k)}$ is the $k$-th derivative of $\varphi$. Assume $\sigma$ is Lipschitz continuous, non-decreasing. Then:*

(a) *$\gamma = 0$ and $\rho = 0$ is a fixed point. Furthermore, assuming that $\sigma$ is non-zero on a set of positive Lebesgue measure, $\varphi$ is non-zero almost everywhere and $\bar{\tau} \in (0, \infty)$, we have if $\rho = 0$, it must be that $\gamma = 0$; in other words, if $\gamma > 0$, then $\rho > 0$.*

(b) *If $\varphi$ is bounded, then $0 \leq \rho \leq C$, and $|\gamma| \leq C / \bar{\tau}$.*

(c) *Given $\gamma = 0$, if $\beta < 1 / \kappa^2$, $\rho = 0$ is a stable fixed point, and if $\beta > 1 / \kappa^2$, $\rho = 0$ is unstable.*

(d) *Consider $\bar{\tau} \in (0, \infty)$. Assume that*

- *$\sigma$ is positive on a set of positive Lebesgue measure, and $\mathbb{E} \left\{ \sigma' \left( \bar{\tau} z \right) \right\} > 0$ for $z \sim \mathcal{N} \left( 0, 1 \right)$,*
- *either*

- *case 1: $\sigma(u) = 0$ for $u \leq 0$, and $\Delta_\rho(u, t) < 0$ for $u, t, \rho > 0$, or*
- *case 2: $\sigma$ is an odd function, and $\Delta_\rho(u, t) < \Delta_\rho(-u, t)$ for $u, t, \rho > 0$,*

*in which $\Delta_\rho(u, t) = \varphi'\left(\beta u + \sqrt{\beta\rho}t\right) - \varphi'\left(\beta u - \sqrt{\beta\rho}t\right)$,*

- *$\varphi$ satisfies $\mathbb{E}\{z\varphi(z)\} \to +\infty$ for $z \sim \mathcal{N}(0, s^2)$ and $s \to \infty$,*
- *$\varphi$ satisfies $\mathbb{E}\{z\mathcal{I}(z)\} > 0$ for $z$ is any Gaussian with zero mean and non-zero variance, and $\mathcal{I}(u) = \varphi(u) + u\varphi'(u)$.*

*Then for any given $\rho > 0$, there exists $\beta^* = \beta^*(\rho, \bar{\tau}) > 0$ finite such that if $\beta \leq \beta^*$, then $\gamma = 0$ is the only fixed point of the equation $\gamma = G(\gamma, \rho)$ and is stable; if $\beta > \beta^*$, $\gamma = 0$ is unstable, and there is one more fixed point at $\gamma > 0$ finite, which is stable.*

(e) *Consider $\bar{\tau} \in (0, \infty)$. The same conclusion as in Claim (c) holds for $\rho = 0$ with $\beta^* = 1/(\kappa\mathbb{E}\{\sigma'(\bar{\tau}z_1)\})$, assuming*

- *$\sigma$ is positive on a set of positive Lebesgue measure, and $\mathbb{E}\{\sigma'(\bar{\tau}z)\} > 0$ for $z \sim \mathcal{N}(0, 1)$,*
- *$\varphi''(u) < 0$ for $u > 0$, and either*
  - *case 1: $\sigma(u) = 0$ for $u \leq 0$, or*
  - *case 2: $\varphi''$ and $\sigma$ are odd functions.*

The assumption $\left\|\varphi^{(k)}\right\|_\infty \leq C$ for $k = 0, ..., 3$ is not critical, only serves to ensure integrability of various terms in the proof and is likely relaxable, but is made for simplicity. The following lemma establishes certain properties of the $\tanh$ function.

**Lemma 12.** *Consider $\varphi = \tanh$. Then:*

(a) *For $\Delta(u, t) = \varphi'(u + t) - \varphi'(u - t)$, we have $\Delta(u, t) < 0$ for $u, t > 0$, and $\Delta(u, t) < \Delta(-u, t)$ for $u, t > 0$.*

(b) *$\lim_{s \to \infty} \mathbb{E}\{z\varphi(z)\} = +\infty$ for $z \sim \mathcal{N}(0, s^2)$.*

(c) *$\mathbb{E}\{z\mathcal{I}(z)\} > 0$ for $z \sim \mathcal{N}(0, s^2)$, $s \neq 0$, and $\mathcal{I}(u) = \varphi(u) + u\varphi'(u)$.*

Now let us consider $\varphi = \sigma = \tanh$. Note that $\tanh'(u) \in (0, 1)$ for any $u \neq 0$, $\tanh'(0) = 1$ and $\mathbb{E}\{\tanh'(\bar{\tau}z_1)\} < 1$ unless $\bar{\tau} = 0$. By Lemma 12, Proposition 11 applies. The following picture is suggested based on Proposition 11:

- When $\beta < 1$, we have convergence to $\gamma = \rho = 0$. This is based on Claim (c) and (e).
- The phase transition for $\rho$ locates at $\beta = 1$, above which we have convergence to $\rho \in (0, 1)$ given a non-zero initialization, and below which $\rho = 0$. Here $\rho < 1$ since $\tanh$ is bounded by 1. This is based on Claim (a), (b), and (c).
- The phase transition of $\gamma$ locates at some $\beta > 1$, above which we have convergence to $\gamma > 0$ given a non-zero initialization, and below which $\gamma = 0$. For $\bar{\tau} > 0$, $\gamma$ cannot grow to $+\infty$ as $\beta$ varies. This is based on Claim (a), (b) and (d).

The proposition also suggests that the two phase transitions are close to each other if $\mathbb{E}\{\sigma'(\bar{\tau}z_1)\} \approx 1$. This requires that $\bar{\tau}^2 \approx 0$, and $\sigma_W^2 \approx 1$ in the case $\alpha = 1$. With respect to Eq. (1), we then have $\sigma_b^2 \approx 0$. Remarkably this is reminiscent of the context of random feedforward networks with $\tanh$ activations, in which (Pennington et al. (2017)) finds that an initialization at $\sigma_W^2 \approx 1$ and $\sigma_b^2 \approx 0$ works better than most edge-of-chaos initialization schemes with Gaussian weights.

We also expect from the proposition a similar picture for $\sigma$ being the ReLU, with a crucial difference. In this case, $\mathbb{E}\{\sigma'(\bar{\tau}z_1)\} = 0.5$, and therefore one cannot have that the two phase transitions being close to each other.

Interestingly Claim (a) implies that the phase transition of $\rho$ never occurs before that of $\gamma$, regardless of the specific $\varphi$ and $\sigma$. One way for the phase transitions to be close to each other is, as above, taking $\varphi = \sigma = \tanh$ and $\bar{\tau}^2 \approx 0$. Claim (a), (c) and (e) of the proposition also suggests that if $\mathbb{E}\{\sigma'(\bar{\tau}z_1)\} > \varphi'(0)$, then $\gamma$ and $\rho$ will share the exact same location of the phase transitions, below which they are zero and above which they are positive.

### C.3 PROOFS FOR SECTION C.2

*Proof of Proposition 10.* Let $\theta = \beta\gamma\sigma(\bar{\tau}z_1) + \sqrt{\beta\rho}z_2$ for brevity.

**Claim (a).** We have $G(0, \rho) = 0$ and $R(0, 0) = 0$. Simple calculations yield $R(0, \rho) = \beta\rho/2$. Claim (a) is then immediate.

**Claim (b).** To study $\rho$ in the case $\gamma \to \infty$, we calculate:

$$
\frac{1}{\rho}R(\gamma, \rho) = \frac{1}{\rho}\mathbb{E}\left\{\int_{z=-\gamma\sigma(\bar{\tau}z_1)\sqrt{\beta/\rho}}^{+\infty}\left(\beta\gamma\sigma(\bar{\tau}z_1) + \sqrt{\beta\rho}z\right)^2\phi(z)\,dz\right\}
$$
$$
= \beta\mathbb{E}\left\{\sqrt{\frac{\beta}{\rho}}\gamma\sigma(\bar{\tau}z_1)\phi\left(\sqrt{\frac{\beta}{\rho}}\gamma\sigma(\bar{\tau}z_1)\right) + \left(1 + \frac{\beta}{\rho}\gamma^2\sigma(\bar{\tau}z_1)^2\right)\Phi\left(\sqrt{\frac{\beta}{\rho}}\gamma\sigma(\bar{\tau}z_1)\right)\right\}
$$

where $\phi(u) = \frac{1}{\sqrt{2\pi}}\exp\{-u^2/2\}$ and $\Phi(u) = \int_{t=-\infty}^u \phi(t)\,dt$ the Gaussian PDF and CDF. Recall that we must have $\frac{1}{\rho}R(\gamma, \rho) = 1$ unless $\rho = \infty$. Also notice that $\phi(\infty) = 0$ and $\Phi(+\infty) = 1$. Since $\sigma$ is positive on a set of positive Lebesgue measure, if $\gamma^2/\rho \to \infty$ as $\gamma \to \infty$, $\frac{1}{\rho}R(\gamma, \rho) \to \infty$, and hence it must be that $\rho \to \infty$ when $\gamma \to \infty$.

**Claim (c).** We compute the first partial derivative $\partial_\gamma G(\gamma, \rho)$, using the fact $\varphi''(u) = \delta(u = 0)$ (in the distributional sense) for the ReLU:

$$
\partial_\gamma G(\gamma, \rho) = \beta\mathbb{E}\{\varphi'(\theta)\sigma'(\bar{\tau}z_1)\} + \beta^2\gamma\mathbb{E}\{\varphi''(\theta)\sigma'(\bar{\tau}z_1)\sigma(\bar{\tau}z_1)\}
$$
$$
= \beta\mathbb{E}\{\varphi'(\theta)\sigma'(\bar{\tau}z_1)\} + \beta^2\gamma\mathbb{E}\{\delta(\theta = 0)\sigma'(\bar{\tau}z_1)\sigma(\bar{\tau}z_1)\}
$$
$$
= \beta\mathbb{E}\{\varphi'(\theta)\sigma'(\bar{\tau}z_1)\} + \beta^2\gamma\mathbb{E}\left\{\frac{1}{\sqrt{2\pi}}\exp\left\{-\frac{\beta\gamma^2\sigma(\bar{\tau}z_1)^2}{2\rho}\right\}\sigma'(\bar{\tau}z_1)\sigma(\bar{\tau}z_1)\right\},
$$

where in the last step, we use the fact that $z_1$ and $z_2$ are independent. Hence some algebras yield the second partial derivative:

$$
\partial_\gamma^2 G(\gamma, \rho) = \sqrt{\frac{2}{\pi}}\beta^2\mathbb{E}\left\{\sigma'(\bar{\tau}z_1)\sigma(\bar{\tau}z_1)\exp\left\{-\frac{\beta\gamma^2\sigma(\bar{\tau}z_1)^2}{2\rho}\right\}\left(1 - \frac{\beta\gamma^2}{2\rho}\sigma(\bar{\tau}z_1)^2\right)\right\}
$$
$$
\overset{(i)}{=} \sqrt{\frac{2}{\pi}}\beta^2\mathbb{E}\{\mathcal{I}'(\sigma(\bar{\tau}z_1))\sigma'(\bar{\tau}z_1)\}
$$
$$
\overset{(ii)}{=} \sqrt{\frac{2}{\pi}}\frac{\beta^2}{\bar{\tau}}\mathbb{E}\{\mathcal{I}(\sigma(\bar{\tau}z_1))z_1\}
$$

where in step $(i)$, we define

$$
\mathcal{I}(u) = \frac{1}{2}u^2\exp\left\{-\frac{\beta\gamma^2}{2\rho}u^2\right\},
$$

in step $(ii)$, we use Stein's lemma. Recall $\bar{\tau} \in (0, \infty)$. Since $\sigma(u) = 0$ for all $u \le 0$, we have $\mathcal{I}(\sigma(\bar{\tau}z)) = 0$ for $z \le 0$. In addition, $\mathcal{I}(u) > 0$ almost everywhere and $\sigma$ is non-zero on a set of positive Lebesgue measure. It follows that $\partial_\gamma^2 G(\gamma, \rho) > 0$. Therefore for any fixed $\rho$, the mapping $\gamma \mapsto G(\gamma, \rho)$ is strictly convex. Notice that $G(0, \rho) = 0$. Hence it cannot be that there is a stable fixed point at $\gamma > 0$.

Next we have:

$$
\partial_\gamma G(+\infty, \rho) = \lim_{\gamma \to +\infty}\beta\mathbb{E}\{\varphi'(\beta\gamma\sigma(\bar{\tau}z_1))\sigma'(\bar{\tau}z_1)\} = \beta\mathbb{E}\{\sigma'(\bar{\tau}z_1)\},
$$
$$
\partial_\gamma G(0, \rho) = \beta\mathbb{E}\left\{\varphi'\left(\sqrt{\beta\rho}z_2\right)\sigma'(\bar{\tau}z_1)\right\} = \frac{1}{2}\beta\mathbb{E}\{\sigma'(\bar{\tau}z_1)\}.
$$

Recall that $\gamma \mapsto G(\gamma, \rho)$ is strictly convex. As such, when $\beta\mathbb{E}\{\sigma'(\bar{\tau}z_1)\} \le 1$, there is only one fixed point at $\gamma = 0$, which is stable. When $\beta\mathbb{E}\{\sigma'(\bar{\tau}z_1)\} \in (1, 2)$, there are two: one at $\gamma = 0$, which is stable, and the other at $\gamma > 0$, which is unstable. When $\beta\mathbb{E}\{\sigma'(\bar{\tau}z_1)\} \ge 2$, there is only one fixed point at $\gamma = 0$, which is unstable.

**Claim (d).** In the case that $\sigma$ is an odd function, $u \mapsto \mathcal{I}(u)$ is even and hence $z \mapsto \mathcal{I}(\sigma(\bar{\tau}z))$ is even. Then from the calculation of Claim (c), it is easy to see that $\partial_\gamma^2 G(\gamma, \rho) = 0$. Also, $G(0, \rho) = 0$. Therefore for each $\rho$, the mapping $\gamma \mapsto G(\gamma, \rho)$ is a straight line which passes through the point $(0, 0)$. Since $\partial_\gamma G(0, \rho) = \frac{1}{2}\beta\mathbb{E}\{\sigma'(\bar{\tau}z_1)\}$, the claim is then immediate.

**Claim (e).** We recall the formula for $R(\gamma, \rho)$ derived in the proof of Claim (b). Since $\sigma$ is odd, $\phi$ is even and $\Phi - 0.5$ is odd, we have

$$\frac{1}{\rho}R(\gamma, \rho) = \beta\mathbb{E}\left\{\sqrt{\frac{\beta}{\rho}}\gamma\sigma(\bar{\tau}z_1)\phi\left(\sqrt{\frac{\beta}{\rho}}\gamma\sigma(\bar{\tau}z_1)\right) + \left(1 + \frac{\beta}{\rho}\gamma^2\sigma(\bar{\tau}z_1)^2\right)\left(\Phi\left(\sqrt{\frac{\beta}{\rho}}\gamma\sigma(\bar{\tau}z_1)\right) - \frac{1}{2}\right)\right\}$$

$$+ \mathbb{E}\left\{\frac{\beta}{2}\left(1 + \frac{\beta}{\rho}\gamma^2\sigma(\bar{\tau}z_1)^2\right)\right\}$$

$$= \mathbb{E}\left\{\frac{\beta}{2}\left(1 + \frac{\beta}{\rho}\gamma^2\sigma(\bar{\tau}z_1)^2\right)\right\}.$$

Then if $\beta > 2$, we always have $R(\gamma, \rho) > \rho$. $\qquad\square$

*Proof of Proposition 11.* Let $\theta = \beta\gamma\sigma(\bar{\tau}z_1) + \sqrt{\beta\rho}z_2$ for brevity.

**Claim (a).** The fact $\gamma = \rho = 0$ is a fixed point is obvious by assumption. Note that in order that $R(\gamma, 0) = 0$, one must have $\mathbb{E}\left\{\varphi(\beta\gamma\sigma(\bar{\tau}z_1))^2\right\} = 0$. By assumption, this requires $\gamma = 0$.

**Claim (b).** The fact that $0 \leq \rho \leq C$ is obvious. Note that

$$|\gamma| = \frac{1}{\bar{\tau}^2}\left|\mathbb{E}\left\{\bar{\tau}z_1\varphi\left(\beta\gamma\sigma(\bar{\tau}z_1) + \sqrt{\beta\rho}z_2\right)\right\}\right| \leq \frac{C}{\bar{\tau}}\mathbb{E}\{|z_1|\} = \frac{C}{\bar{\tau}}\sqrt{\frac{2}{\pi}},$$

which yields the claim.

**Claim (c).** We have:

$$\partial_\rho R(\gamma, \rho) = \sqrt{\frac{\beta}{\rho}}\mathbb{E}\{\varphi(\theta)\varphi'(\theta)z_2\} \stackrel{(i)}{=} \beta\mathbb{E}\left\{\varphi'(\theta)^2 + \varphi''(\theta)\varphi(\theta)\right\},$$

where $(i)$ is due to Stein's lemma. As such, $\partial_\rho R(0, 0) = \beta\kappa^2$. The claim is then immediate.

**Claim (d).** Consider $\rho > 0$. We have:

$$\partial_\gamma G(\gamma, \rho) = \beta\mathbb{E}\{\varphi'(\theta)\sigma'(\bar{\tau}z_1)\} + \beta^2\gamma\mathbb{E}\{\varphi''(\theta)\sigma'(\bar{\tau}z_1)\sigma(\bar{\tau}z_1)\},$$

and therefore,

$$\partial_\gamma^2 G(\gamma, \rho) = 2\beta^2\mathbb{E}\{\varphi''(\theta)\sigma'(\bar{\tau}z_1)\sigma(\bar{\tau}z_1)\} + \beta^3\gamma\mathbb{E}\left\{\varphi'''(\theta)\sigma'(\bar{\tau}z_1)\sigma(\bar{\tau}z_1)^2\right\}$$

$$= \frac{\beta^2}{\bar{\tau}}\mathbb{E}\left\{\partial_{z_1}\left(\varphi''\left(\beta\gamma\sigma(\bar{\tau}z_1) + \sqrt{\beta\rho}z_2\right)\sigma(\bar{\tau}z_1)^2\right)\right\}$$

$$= \frac{\beta^2}{\bar{\tau}}\mathbb{E}_{z_2}\left\{\mathbb{E}_{z_1}\left\{\varphi''\left(\beta\gamma\sigma(\bar{\tau}z_1) + \sqrt{\beta\rho}z_2\right)z_1\sigma(\bar{\tau}z_1)^2\right\}\right\}$$

$$= \frac{\beta^2}{\bar{\tau}}\mathbb{E}_{z_1}\left\{\mathbb{E}_{z_2}\left\{\varphi''\left(\beta\gamma\sigma(\bar{\tau}z_1) + \sqrt{\beta\rho}z_2\right)\right\}z_1\sigma(\bar{\tau}z_1)^2\right\}$$

where we use Stein's lemma and the fact $z_1$ and $z_2$ are independent. Notice that, for $\beta\rho > 0$, by Stein's lemma,

$$\mathbb{E}_{z_2}\left\{\varphi''\left(a + \sqrt{\beta\rho}z_2\right)\right\} = \frac{1}{\sqrt{\beta\rho}}\mathbb{E}_{z_2}\left\{z_2\varphi'\left(a + \sqrt{\beta\rho}z_2\right)\right\}$$

$$= \frac{1}{\sqrt{2\pi\beta\rho}}\int_{t=0}^{+\infty}\left[\varphi'\left(a + \sqrt{\beta\rho}t\right) - \varphi'\left(a - \sqrt{\beta\rho}t\right)\right]t\exp\left\{-\frac{t^2}{2}\right\}dt.$$

Consider case 1. Since $\sigma(u) = 0$ for all $u \leq 0$,

$$\partial_\gamma^2 G(\gamma, \rho) = \frac{\beta^2}{2\pi\bar{\tau}\sqrt{\beta\rho}}\int_{z=0}^{+\infty}\int_{t=0}^{+\infty}\Delta_\rho(\gamma\sigma(\bar{\tau}z), t)tz\sigma(\bar{\tau}z)^2\exp\left\{-\frac{z^2 + t^2}{2}\right\}dtdz.$$

Since $\Delta_\rho(u,t) < 0$ for any $u > 0$ and $t > 0$, and by the assumption that $\sigma$ is positive on a set of positive Lebesgue measure (which cannot intersect with $(-\infty; 0]$ since $\sigma(u) = 0$ for all $u \leq 0$), we have $\partial_\gamma^2 G(\gamma, \rho) < 0$ for $\gamma > 0$. In case 2, since $\sigma$ is an odd function,

$$\partial_\gamma^2 G(\gamma, \rho) = \frac{\beta^2}{2\pi\bar{\tau}\sqrt{\beta\rho}} \int_{z=0}^{+\infty} \int_{t=0}^{+\infty} [\Delta_\rho(\gamma\sigma(\bar{\tau}z), t) - \Delta_\rho(-\gamma\sigma(\bar{\tau}z), t)] tz\sigma(\bar{\tau}z)^2 \exp\left\{-\frac{z^2+t^2}{2}\right\} dtdz.$$

Since $\Delta_\rho(u,t) < \Delta_\rho(-u,t)$ for $u > 0$ and $t > 0$ and $\sigma(u) \geq 0$ for $u \geq 0$ (for $\sigma$ being odd and non-decreasing), we again have $\partial_\gamma^2 G(\gamma, \rho) < 0$ for $\gamma > 0$. As such, the mapping $\gamma \mapsto G(\gamma, \rho)$, for $\rho > 0$, is strictly concave on $(0, \infty)$.

Next we have

$$\partial_\gamma G(0, \rho) = \beta\mathbb{E}\left\{\varphi'\left(\sqrt{\beta\rho}z_2\right)\sigma'(\bar{\tau}z_1)\right\} = \beta\mathbb{E}\left\{\varphi'\left(\sqrt{\beta\rho}z_2\right)\right\}\mathbb{E}\left\{\sigma'(\bar{\tau}z_1)\right\},$$

since $z_1$ and $z_2$ are independent. Notice that by Stein's lemma,

$$\partial_\beta\left(\beta\mathbb{E}\left\{\varphi'\left(\sqrt{\beta\rho}z_2\right)\right\}\right) = \frac{1}{2}\mathbb{E}\left\{\mathcal{I}'\left(\sqrt{\beta\rho}z_2\right)\right\} = \frac{1}{2\beta\rho}\mathbb{E}\left\{\sqrt{\beta\rho}z_2\mathcal{I}\left(\sqrt{\beta\rho}z_2\right)\right\}.$$

By assumption, we thus have that $\beta\mathbb{E}\left\{\varphi'\left(\sqrt{\beta\rho}z_1\right)\right\}$ and hence $\partial_\gamma G(0, \rho)$ are increasing in $\beta$. Furthermore, $\partial_\gamma G(0, \rho) \to 0$ when $\beta \to 0$, and

$$\lim_{\beta\to\infty} \partial_\gamma G(0, \rho) \overset{(i)}{=} \mathbb{E}\left\{\sigma'(\bar{\tau}z_1)\right\}\lim_{\beta\to\infty}\frac{1}{\rho}\mathbb{E}\left\{\sqrt{\beta\rho}z_2\varphi\left(\sqrt{\beta\rho}z_2\right)\right\} \overset{(ii)}{=} +\infty$$

where $(i)$ is by Stein's lemma, and $(ii)$ is by assumption. Therefore, there must exist $\beta^* = \beta^*(\rho, \bar{\tau}) > 0$ finite such that $\partial_\gamma G(0, \rho) = 1$ at $\beta = \beta^*$, $\partial_\gamma G(0, \rho) < 1$ for $\beta < \beta^*$ and $\partial_\gamma G(0, \rho) > 1$ for $\beta > \beta^*$. Since $\gamma \mapsto G(\gamma, \rho)$ is strictly concave on $(0, \infty)$ and $\gamma = 0$ is a fixed point by Claim (a), $\beta^*$ is the threshold as in the claim.

**Claim (e).** Consider $\rho = 0$, in which case:

$$\partial_\gamma^2 G(\gamma, 0) = \frac{\beta^2}{\bar{\tau}\sqrt{2\pi}} \int_{z=-\infty}^{+\infty} \varphi''(\beta\gamma\sigma(\bar{\tau}z)) z\sigma(\bar{\tau}z)^2 \exp\left\{-\frac{z^2}{2}\right\} dz.$$

Consider case 1. Since $\sigma(u) = 0$ for all $u \leq 0$,

$$\partial_\gamma^2 G(\gamma, 0) = \frac{\beta^2}{\bar{\tau}\sqrt{2\pi}} \int_{z=0}^{+\infty} \varphi''(\beta\gamma\sigma(\bar{\tau}z)) z\sigma(\bar{\tau}z)^2 \exp\left\{-\frac{z^2}{2}\right\} dz.$$

Since $\varphi''(u) < 0$ for $u > 0$ and $\sigma$ is positive on a set of positive Lebesgue measure (which cannot intersect $(-\infty; 0]$ since $\sigma(u) = 0$ for all $u \leq 0$), it is easy to see that $\partial_\gamma^2 G(\gamma, 0) < 0$ for $\gamma > 0$. In case 2, since $\varphi''$ and $\sigma$ are odd, the mapping $z \mapsto \varphi''(\beta\gamma\sigma(\bar{\tau}z))$ is odd, and hence

$$\partial_\gamma^2 G(\gamma, 0) = \frac{2\beta^2}{\bar{\tau}\sqrt{2\pi}} \int_{z=0}^{+\infty} \varphi''(\beta\gamma\sigma(\bar{\tau}z)) z\sigma(\bar{\tau}z)^2 \exp\left\{-\frac{z^2}{2}\right\} dz.$$

Again we have $\partial_\gamma^2 G(\gamma, 0) < 0$ for $\gamma > 0$. As such, $\gamma \mapsto G(\gamma, \rho)$ is strictly concave on $(0, \infty)$. Notice that $\partial_\gamma G(0, 0) = \beta\kappa\mathbb{E}\left\{\sigma'(\bar{\tau}z_1)\right\}$, linearly increasing in $\beta$. This proves the claim with $\beta^* = 1/(\kappa\mathbb{E}\left\{\sigma'(\bar{\tau}z_1)\right\})$. $\qquad\square$

*Proof of Lemma 12.* We prove the first claim:

$$\Delta_\rho(u, t) = \tanh^2(u - t) - \tanh^2(u + t).$$

Since $\tanh^2$ is even and increasing on $(0, +\infty)$, $\Delta_\rho(u, t) < 0$ for $u, t > 0$ and $\Delta_\rho(u, t) > 0$ for $u > 0$ and $t < 0$. Simple algebra yields

$$\Delta_\rho(-u, t) - \Delta_\rho(u, t) = 2\Delta_\rho(u, -t).$$

Hence $\Delta_\rho(u, t) < \Delta_\rho(-u, t)$ for $u > 0$ and $t > 0$. To see the second claim, for $z \sim \mathcal{N}(0, 1)$,

$$\lim_{s\to\infty} \mathbb{E}\left\{sz\varphi(sz)\right\} = \lim_{s\to\infty}\mathbb{E}\left\{s|z|\right\} = +\infty.$$

To see the third claim, since $\mathcal{I}$ is odd,

$$\mathbb{E}\left\{z\mathcal{I}(z)\right\} = 2\mathbb{E}\left\{z\left(\tanh(z) + z\left(1 - \tanh(z)^2\right)\right)\mathbb{I}(z > 0)\right\} > 0.$$

$$\qquad\square$$

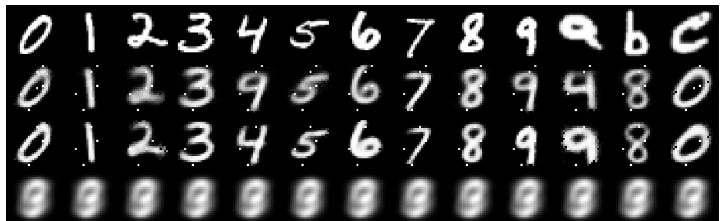

Figure 10: The reconstructions by the schemes from Table 1, as described in Appendix D.1, in Setting 1 (i.e., $\varphi_0 = \tanh$). From the top row: original images, reconstructions from Scheme 1, 4 and 2. We omit the reconstructions from other schemes, since they are almost identical to those of Scheme 2. For each digit/letter category, the image is selected from the test set by ranking the reconstruction loss, averaged across Scheme 1 and 4, and picking one at the 75% percentile.

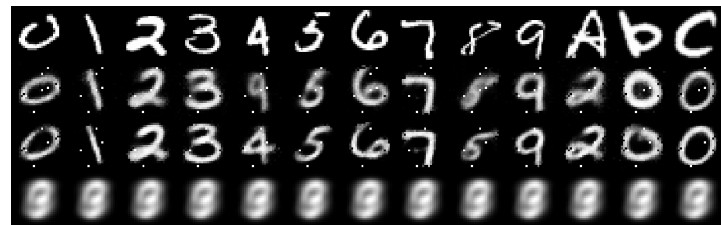

Figure 11: Description similar to Fig. 10. The chosen images are the 25% percentile.

## D  VISUALIZATION FOR SECTION 3.5

### D.1  VISUALIZATION OF THE RECONSTRUCTIONS

In Fig. 10, 11, 12 and 13, we show the reconstructions of several images by the trained networks after $5 \times 10^5$ training iterations, under the schemes from Table 1, in the experiments of Section 3.5. We draw 10 digit images from the MNIST test set, as well as 3 letter images from the EMNIST Letters test set (Cohen et al. (2017)). Note that the networks are not trained with any letter images from the EMNIST data set. The reconstruction quality is visually imperfect even after intensive training, which is entirely expected for vanilla autoencoders and regression problems.

Observe that for the schemes that yield meaningful reconstructions, they output recognizable digits for digit images, while for letter images, most of their reconstructions are hardly recognizable as letters. As such, the trained networks of these schemes do not simply approximate the identity function, but rather capture some low-dimensional structures of the data. An exception is Scheme 7 under Setting 2, which is not surprising since it is a purely $\tanh$ network and $\tanh$ is almost identity near zero.

### D.2  VISUALIZATION OF THE EVOLUTION

We show the evolution of the test reconstruction loss on the plane $\left(\sigma_W^2, \sigma_b^2\right)$, in conjunction with the experiments of Section 3.5. To make the computation more manageable, we opt for $L = 50$ with less iterations, while maintaining other parameters the same as in Section 3.5. The results are shown in Fig. 14 and 15. Several patterns emerge in good agreement with our hypotheses. Firstly, for $\varphi = \mathrm{ReLU}$, the evolution starts earliest near $\sigma_W^2 = 2$, shows almost no progress or is numerically unstable when $\sigma_W^2 \gg 2$, and is much slower when $\sigma_W^2 \ll 2$. Secondly, for $\varphi = \tanh$, the evolution is much slower when $\sigma_W^2 \ll 1$. Intriguingly the evolution is almost insensitive to $\sigma_b^2$.

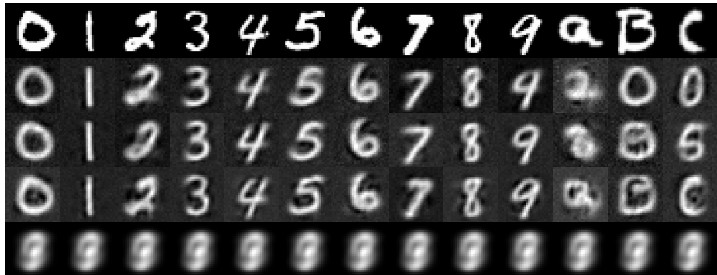

Figure 12: Description similar to Fig. 10. The setting is Setting 2 (i.e., identity $\varphi_0$). The reconstructions are of Scheme 1, 4, 7 and 2. The chosen images are the 75% percentile. The ranking is by averaging over Scheme 1, 4, and 7.

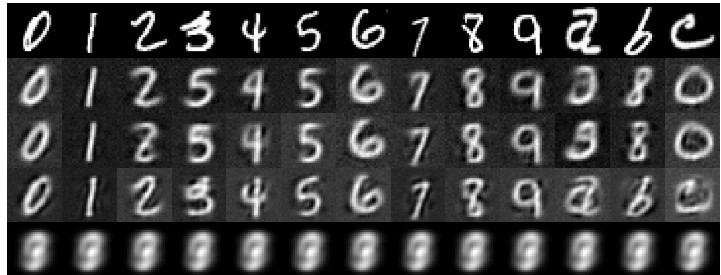

Figure 13: Description similar to Fig. 10. The setting is Setting 2 (i.e., identity $\varphi_0$). The reconstructions are of Scheme 1, 4, 7 and 2. The chosen images are the 25% percentile. The ranking is by averaging over Scheme 1, 4, and 7.

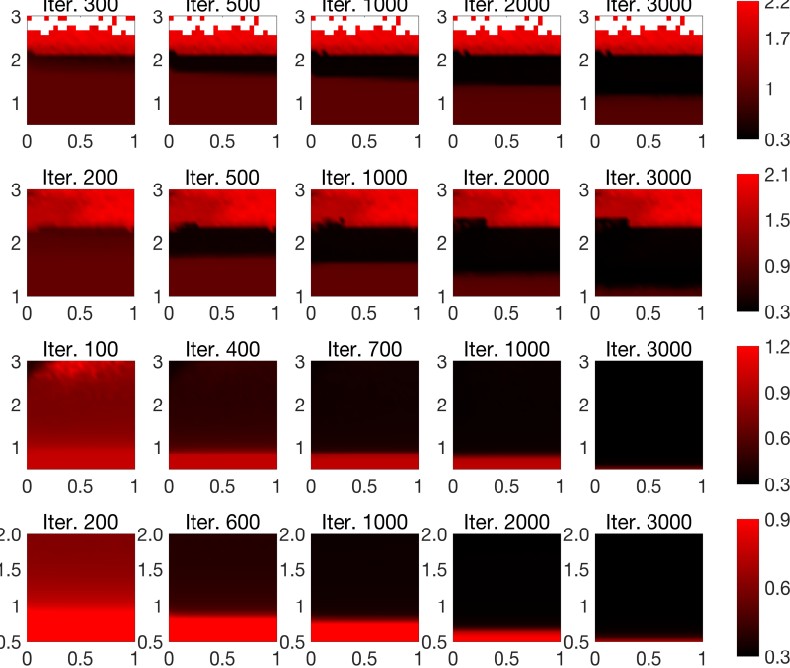

Figure 14: The test loss for various pairs $\left(\sigma_W^2, \sigma_b^2\right)$ at different iterations, as described in Appendix D.2, in Setting 1 (i.e., $\varphi_0 = \tanh$). The horizontal axis is $\sigma_b^2$, and the vertical axis is $\sigma_W^2$. First row: $\varphi = \sigma = \text{ReLU}$; second row: $\varphi = \text{ReLU}$ and $\sigma = \tanh$; third row: $\varphi = \sigma = \tanh$; fourth row: $\varphi = \tanh$ and $\sigma = \text{ReLU}$. Red indicates higher loss, and black indicates lower loss. White indicates a numerical error.

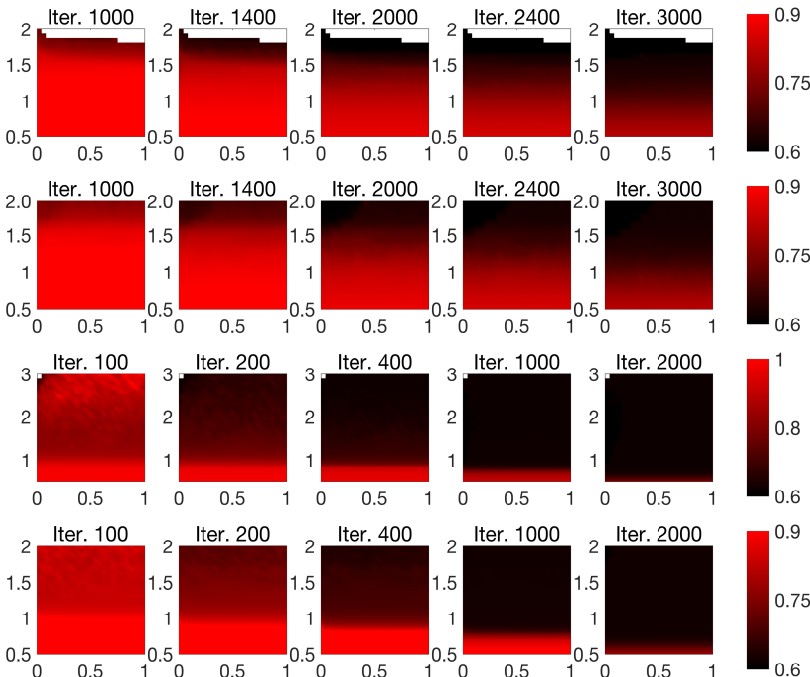

Figure 15: The test loss for various pairs $\left(\sigma_W^2, \sigma_b^2\right)$ at different iterations, as described in Appendix D.2, in Setting 2 (i.e., $\varphi_0$ is the identity). The description is similar to Fig. 14. White indicates either a very large value or a numerical error, which are due to the fact that the chosen learning rate is not sufficiently small. Note that in this setting with $\varphi = \text{ReLU}$, the training process is trapped in numerical errors for $\sigma_W^2 > 2$ as we expect, and hence the test loss at $\sigma_W^2 > 2$ is not plotted.

# E   MISCELLANIES

## E.1   EDGE OF CHAOS INITIALIZATION

We quickly review the edge of chaos (EOC) initialization (Schoenholz et al. (2016)). For an activation $\sigma$, any $\left(\sigma_W^2, \sigma_b^2\right)$ such that there exists a finite $\bar{\tau} \geq 0$ for which

$$\bar{\tau}^2 = \sigma_W^2 \mathbb{E}\left\{\sigma\left(\bar{\tau}z\right)^2\right\} + \sigma_b^2,$$
$$\sigma_W^2 = 1/\mathbb{E}\left\{\sigma'\left(\bar{\tau}z\right)^2\right\}$$

is said to be an EOC initialization scheme, where $z \sim \mathcal{N}(0, 1)$. This is based on the order-to-chaos phenomenon found in (Poole et al. (2016)). For ReLU $\sigma$, $\sigma_W^2 = 2$ and $\sigma_b^2 = 0$ is the only EOC initialization (Hayou et al. (2018)) and coincides with the He initialization (He et al. (2015)). For $\sigma = \tanh$, there can be multiple pairs that form EOC initialization. (Pennington et al. (2017)) argues that a better EOC scheme should be closer to dynamical isometry and suggests taking $\sigma_W^2 \approx 1.05$ and $\sigma_b^2 \approx 2.01 \times 10^{-5}$ for $\sigma = \tanh$. The EOC initialization is applicable to feedforward networks.

## E.2   USEFUL FACTS

We state several useful facts, which are used in various places throughout. First, for a Gaussian matrix $\boldsymbol{W}$ and an independent vector $\boldsymbol{u}$, we have $\boldsymbol{W}\boldsymbol{u}$ is also Gaussian. In particular, if entries of $\boldsymbol{W}$ are i.i.d. $\mathcal{N}\left(0, s^2\right)$, then $\boldsymbol{W}\boldsymbol{u} \sim \mathcal{N}\left(0, s^2 \|\boldsymbol{u}\|^2 \boldsymbol{I}\right)$. The second fact is Stein's lemma:

**Lemma 13** (Stein's). $\mathbb{E}\left\{zf\left(z\right)\right\} = \mathbb{E}\left\{f'\left(z\right)\right\}$ *for* $z \sim \mathcal{N}(0, 1)$ *and* $f : \mathbb{R} \mapsto \mathbb{R}$ *weakly differentiable whenever the expectations are defined.*

