# OpenReview forum: "On Random Deep Weight-Tied Autoencoders: Exact Asymptotic Analysis, Phase Transitions, and Implications to Training"
_ICLR.cc/2019/Conference_

### Official Review · AnonReviewer2 · 2018-10-28
**An intriguing work bringing non-trivial analytical insights on the behaviour of deep autoencoders with random tied weights. Likely to generate more work in this direction.**

**Rating:** 8
**Confidence:** 4

**Review:**

This paper studies auto-encoders under several assumptions: (a) the auto-encoder's layers are fully connected, with random weights, (b) the auto-encoder is weight-tied, (c) the dimensions of the layers go to infinity with fixed ratios. The main contribution of the paper is to point out that this model of random autoencoder can be elegantly and rigorously analysed with one-dimensional equations. The idea is original and will probably lead to new directions of research. Already the first applications that the paper suggests are exciting.

The paper does a good job in justifying assumptions (a), (b) and (c) in the introduction. It is convincing in the fact that this point of view may bring practical insights on training initialization for real-world autoencoders. Thus my opinion is that this paper brings original and significant ideas in the field.

One flaw of this paper is that the writing might be clearer. For instance when presenting the technical theorem (Theorem 1), it would be useful to have an intuitive explanation for the theorem and the state-evolution-like equations. However, I believe that there are some easy fixes that would greatly improve the clarity of the exposition. Here is a list of suggestions:

- In Section 2.1, a large number of notations are introduced. It would help a lot if the authors made a graphical representation of these. For instance, a diagram where every linearity / non-linearity is a box, and the different variables $x_l$, $\hat{x}_l$ appear would help a lot.

- Section 2.2 is rather technical. The authors could try to give some more intuition of what's happening. For instance, they could spend more time after the theorem explaining what $\tau_l, \gamma_l$ and $\rho_l$ mean. They could also introduce the notation S_sig and S_var early and this section (and not in Section 3), because it helps interpreting the parameters. It would also help if they could write a heuristic derivation of the state-evolution-like equations. From the paper, the only way the reader can understand the intuition behind those complicated equations is to look at the proof of Theorem 1 (which is rather technical).

- In Section 3.1, I did not understand the difference between interpretations 1 and 2. Could the authors clarify?

- In Section 3.4, I did not understand the sentence: "In particular, near the phase transition of \gamma, S_sig/S_var = \Omega(\beta^{1.5}". If one uses the \Omega notation, it means that some parameter is converging to something. What is the parameter? As a consequence, I did not understand this paragraph.

- In Section 3.5, the authors should make clear from the beginning why they are running those specific simulations. What hypothesis are they trying to check? I finally concluded that they are running simulations to check if the hypothesis they make in the first paragraph are true. They also want to compare with some other criteria in the literature, named EOC, that also gives insights about the trainability of the network. However, they could explicitly say in the beginning of the second paragraph that this is the goal.

- In a similar spirit, the authors should end Section 3.5 with a clear conclusion on whether or not the framework enables us to predict the trainability of the autoencoder.



Minor edits / remarks:

- Typo: last but one paragraph of the introduction: "whose analysis is typically more straighforwards" -> "straightforward".

- At the end of Section 3.2: what can be proved about the behavior of \gamma / \sqrt{\rho}? It is obviously a central quantity and the authors do not say what happens in the phases where \gamma and \rho go to infinity for instance. Is it because it is hard to analyse?

---

> ### Author Response · Authors · 2018-11-26
> **Thank you for helpful suggestions. Response Part 1/2**
>
> We very much appreciate that you find our work interesting, as well as your suggestions to improve readability. We also agree that the analysis here has the potentially to lead to more exciting research directions.
>
> We provide specific reply to each comment in the following.
>
>
> ********************************************
> Question: “In Section 2.1, a large number of notations are introduced. It would help a lot if the authors made a graphical representation of these. For instance, a diagram where every linearity / non-linearity is a box, and the different variables $x_l$, $\hat{x}_l$ appear would help a lot.”
>
> Answer: Thank you for the suggestion. We update with a schematic diagram in Figure 5 in the appendix. Admittedly we could not find a way to fit the diagram within the main 10 pages.
>
>
> ********************************************
> Question: “Section 2.2 is rather technical. The authors could try to give some more intuition of what's happening. For instance, they could spend more time after the theorem explaining what $\tau_l, \gamma_l$ and $\rho_l$ mean. They could also introduce the notation S_sig and S_var early and this section (and not in Section 3), because it helps interpreting the parameters. It would also help if they could write a heuristic derivation of the state-evolution-like equations. From the paper, the only way the reader can understand the intuition behind those complicated equations is to look at the proof of Theorem 1 (which is rather technical).”
>
> Answer: We thank you for the suggestions. We agree that this is quite technical. In Appendix A.2, we give an outline of the proof and high-level descriptions of the ideas, from which the meanings of \gamma_l, \rho_l, and the intuition behind the state-evolution-like equations are clearer. The reason that we have to defer this task to the appendix is that the task necessarily requires stating a result that is in line with Proposition 5 and Corollary 6. By stating such a result, we face a risk of creating a distraction to many readers of ICLR, while we want to focus more on the implications of Theorem 1 within the main 10 pages.
>
> Nevertheless we have added a sort paragraph at the beginning of Section 2.2, performing heuristic calculations on a special case to motivate the result in Theorem 1. We hope this will make the theorem easier to interpret.
>
> Regarding the notation S_sig and S_var, we did not introduce them earlier since in our opinions, they are not as central as \tau_l, \gamma_l and \rho_l, whose one-dimensional evolutions inherit nice visualizable properties (as shown in Fig 2) that make the phase transition phenomena more intuitive.
>
>
> ********************************************
> Question: “In Section 3.1, I did not understand the difference between interpretations 1 and 2. Could the authors clarify?”
>
> Answer: The two interpretations are closely related. Interpretation 1 states that for most intermediate layers (i.e. 1<< \ell << L), the decoder outputs \hat{x}_\ell can be described with \gamma and \rho. In Interpretation 2, we extend this property to outermost layers (e.g. \ell = 1), which now include the final output of the autoencoder. This requires an extra mild assumption on the normalization of the input x.
>
>
> ********************************************
> Question: “In Section 3.4, I did not understand the sentence: "In particular, near the phase transition of \gamma, S_sig/S_var = \Omega(\beta^{1.5}". If one uses the \Omega notation, it means that some parameter is converging to something. What is the parameter? As a consequence, I did not understand this paragraph.”
>
> Answer: We thank you for pointing out this. We include a footnote, saying by f(\beta)=\Omega(g(\beta)) in a certain range of \beta, we mean d[f(\beta)/g(\beta)]/d\beta>0 on this range. We believe this better reflects the sensitive-to-perturbation behavior. We also remove the use of the big \Theta notation in the previous paragraph, and instead use the proportional relation.

---

> ### Author Response · Authors · 2018-11-26
> **Thank you for helpful suggestions. Response Part 2/2**
>
> Question: “In Section 3.5, the authors should make clear from the beginning why they are running those specific simulations. What hypothesis are they trying to check? I finally concluded that they are running simulations to check if the hypothesis they make in the first paragraph are true. They also want to compare with some other criteria in the literature, named EOC, that also gives insights about the trainability of the network. However, they could explicitly say in the beginning of the second paragraph that this is the goal.
>
> In a similar spirit, the authors should end Section 3.5 with a clear conclusion on whether or not the framework enables us to predict the trainability of the autoencoder. “
>
> Answer: Thank you for the comment. Your understanding is correct. We have made efforts to restructure this section. In particular, we include some headings to signal the readers what to expect. We also refine the descriptions to make the goals and the conclusions clearer.
>
>
> ********************************************
> Question: “Typo: last but one paragraph of the introduction: "whose analysis is typically more straighforwards" -> "straightforward”.”
>
> Answer: This is fixed. Thanks!
>
>
> ********************************************
> Question: “At the end of Section 3.2: what can be proved about the behavior of \gamma / \sqrt{\rho}? It is obviously a central quantity and the authors do not say what happens in the phases where \gamma and \rho go to infinity for instance. Is it because it is hard to analyse?”
>
> Answer: It is indeed hard to prove non-trivial statements on this ratio. We could make a few in the case the decoder activation \varphi is ReLU, but it is trivial. The case \varphi is tanh is much more difficult (note that in this case, \varphi and \rho are bounded).
>
> The proven statements for \gamma and \rho are only intended to verify that we are not missing interesting behaviors by doing numerical simulations. In general, their behaviors, including the ratio \gamma/\sqrt{\rho}, can be conveniently simulated, so we place less emphasis on having rigorous proofs for the observations we make.

---

### Official Review · AnonReviewer1 · 2018-11-02
**Through analysis on weight-tied autoencoders may benefit from more clear presentation**

**Rating:** 8
**Confidence:** 4

**Review:**

This work applies infinite width limit random network framework (a.k.a. Mean field analysis) to study deep autoencoders when weights are tied between encoder and decoder. Random network analysis allows to have exact analysis of asymptotic behaviour where the network is infinitely deep (but width taken to infinite first). This exact analysis allows to answer some theoretical questions from previous works to varying degrees of success.

Building on the techniques from Poole et al (2016) [1], Schoenholz et al (2017) [2], the theoretical analysis to deep autoencoder with weight tied encoder/decoder shows interesting properties. The fact that the network component are split into encoder/decoder architecture choice along with weight tying shows various interesting phase of network configuration.

Main concern with this work is applicability of the theoretical analysis to real networks. The autoencoding samples on MNIST provided in the Appendix at least visually do not seem to be a competitive autoencoder (e.g. blurry and irrelevant pixels showing up).

Also the empirical study with various schemes is little hard to parse and digest. It would be better to restructure this section so that the messages from theoretical analysis in the earlier section can be clearly seen in the experiments.

The experiments done on fixed learning rate should not be compare to other architectures in terms of training speed as learning rates are sensitive to the architecture choice and speed may be not directly comparable.

Questions/Comments
- Without weight tying the whole study is not much different from just the feedforward networks. However, as noted by the authors Vincent et al (2010) showed that empirically autoencoders with or without weight tying performs comparably. What is the benefit of analyzing more complicated case where we do not get a clear benefit from?

- Many auto encoding networks benefit from either bottleneck or varying the widths. The author’s regime is when all of the hidden layers grows to infinity at the same order. Would this limit capture interesting properties of autoencoders?

- When analysis is for weight tied networks, why is encoder and decoder assume to have different non-linearity? It does show interesting analysis but is it a practical choice? From this work, would you recommend using different non-linearities?

- It would be interesting to see how this analysis is applied to Denoising Autoencoders [3], which should be straightforward to apply similar to dropout analysis appeared in Schoenholz et al [2].

[1] Ben Poole, Subhaneil Lahiri, Maithra Raghu, Jascha Sohl-Dickstein, and Surya Ganguli. Exponential
expressivity in deep neural networks through transient chaos. In Advances in neural information
processing systems, pp. 3360–3368, 2016.
[2] S.S. Schoenholz, J. Gilmer, S. Ganguli, and J. Sohl-Dickstein. Deep information propagation. 5th International Conference on Learning Representations, 2017.
[3] Pascal Vincent, Hugo Larochelle, Isabelle Lajoie, Yoshua Bengio, and Pierre-Antoine Manzagol. Stacked denoising autoencoders: Learning useful representations in a deep network with a local denoising criterion. Journal of Machine Learning Research, 11(Dec):3371–3408, 2010.

---

> ### Author Response · Authors · 2018-11-26
> **Thank you for insightful comments. Response Part 1/2.**
>
> We would like to thank the reviewer for providing thoughtful comments. We reply to each comment in the following.
>
> ********************************************
> Question: “Building on the techniques from Poole et al (2016) [1], Schoenholz et al (2017) [2], the theoretical analysis to deep autoencoder with weight tied encoder/decoder shows interesting properties.”
>
> Answer: We agree that our setting is very related to the recent works by Poole, Schoenholz, Pennington, Ganguli and many others in that we consider random weights. On the other hand, we would like to emphasize a key difference: our analysis for weight-tied case heavily depends on the Gaussian conditioning technique from TAP theory / approximate message passing (AMP) literature. The analysis is challenging due to the weight-tied constraint, as acknowledged by previous works (Arora et al 2015, Chen et al 2018). In fact, the work Chen et al 2018 follows the same mean-field framework as Poole et al (2016), Schoenholz et al (2017), but has to assume weight-untied analysis for weight-tied structures. Admittedly we choose not to detail how crucial the use of this technique is in the main 10 pages, and only mention this fact briefly at the end of the second last paragraph of Section 1. This is because doing so would be technically involved, probably lengthy, and will be a distraction to many ICLR readers. We leave that to the appendix for theoretically inclined readers.
>
>
> ********************************************
> Question: “Main concern with this work is applicability of the theoretical analysis to real networks. The autoencoding samples on MNIST provided in the Appendix at least visually do not seem to be a competitive autoencoder (e.g. blurry and irrelevant pixels showing up).”
>
> Answer: We acknowledge this valid point. We focus solely on the effect of depth and analytical tractability, and so a simple setup (vanilla autoencoders) is much desired. This is analogous to the setting in Schoenholtz et al 2017 which considered vanilla feedforward networks - with a key difference that we consider weight-tied structures. It would be very interesting to extend the analysis to more complex setups. Nevertheless this simple setup is sufficiently challenging, yet already yielding interesting implications, considering the fact that this is the first work on weight-tied autoencoders in the considered directions.
>
>
> ********************************************
> Question: “Also the empirical study with various schemes is little hard to parse and digest. It would be better to restructure this section so that the messages from theoretical analysis in the earlier section can be clearly seen in the experiments.”
>
> Answer: We thank you for raising this point. We have made efforts to restructure this section. In particular, we include some headings to signal the readers what to expect. We also refine the descriptions to make the goals and the conclusions clearer.
>
>
> ********************************************
> Question: “The experiments done on fixed learning rate should not be compare to other architectures in terms of training speed as learning rates are sensitive to the architecture choice and speed may be not directly comparable.”
>
> Answer: We would like to clarify that the comparison is made for different initialization schemes (in Table 1) of the same pair of encoder-decoder activations and hence the same architecture.  Since the architecture is the same for each pair, we believe this is a fair comparison. We have added a sentence in the revision to remind readers about this. Thanks.
>
>
> ********************************************
> References:
> Sanjeev Arora, Yingyu Liang, and Tengyu Ma. Why are deep nets reversible: A simple theory, with implications for training. arXiv preprint arXiv:1511.05653, 2015.
>
> Minmin Chen, Jeffrey Pennington, and Samuel S Schoenholz. Dynamical isometry and a mean field theory of rnns: Gating enables signal propagation in recurrent neural networks. arXiv preprint arXiv:1806.05394, 2018.
>
> S.S. Schoenholz, J. Gilmer, S. Ganguli, and J. Sohl-Dickstein. Deep information propagation. 5th International Conference on Learning Representations, 2017.

---

> > ### Comment · AnonReviewer1 · 2018-12-03
> > **Appreciate the response from the authors**
> >
> > Thank you for responding to the comments and concerns.
> >
> > I appreciate the contribution of the paper more now and has reflect that in increased score.

---

> > > ### Author Response · Authors · 2018-12-04
> > > **Thank you.**
> > >
> > > Thank you for your reply. We are happy to know that.

---

> ### Author Response · Authors · 2018-11-26
> **Thank you for insightful comments. Response Part 2/2.**
>
> Question: “Without weight tying the whole study is not much different from just the feedforward networks. However, as noted by the authors Vincent et al (2010) showed that empirically autoencoders with or without weight tying performs comparably. What is the benefit of analyzing more complicated case where we do not get a clear benefit from?”
>
> Answer: We thank the reviewer for raising this important remark by Vincent et al (2010). Since this remark, the weight-tied autoencoder has become standard. Many subsequent experimental works use it. Interestingly all recent theoretical works concerning directly autoencoders also assume weight-tying. In fact, experiments in the paper Vincent et al (2010) are done on weight-tied autoencoders. As such, we believe it is imperative to thoroughly investigate this weight-tied architecture.
>
> We also would like to to emphasize that the weight-tying assumption is very critical to the “approximate inference” notion and the concept of reversibility (which we consider in Section 3.3). Without it, there would be no signal component (i.e. S_sig=0) for any activation and any depth, and so the random autoencoder without weight-tying cannot be said to perform approximate inference.
>
>
> ********************************************
> Question: “Many auto encoding networks benefit from either bottleneck or varying the widths. The author’s regime is when all of the hidden layers grows to infinity at the same order. Would this limit capture interesting properties of autoencoders?”
>
> Answer: While we require that all dimensions go to infinity, we allow their ratios to be arbitrary constants in the main theoretical result Theorem 1. For instance, an autoencoder with dimensions 1000 - 100 - 1000 would fit in this description, in that the dimensions are large, with the dimension ratio being 0.1. Likewise, in the infinite-depth analysis, we also allow the ratio \alpha to be an arbitrary constant.
>
>
> ********************************************
> Question: “When analysis is for weight tied networks, why is encoder and decoder assume to have different non-linearity? It does show interesting analysis but is it a practical choice? From this work, would you recommend using different non-linearities?”
>
> Answer: For the purpose of this work, the case of different encoder/decoder activations is a test case for our hypothesis, which does not place any restriction on whether the activations are identical. It makes an interesting test case because having different activations is not yet considered in the theories for feedforward networks, but might be natural for the autoencoder setup.
>
> On the other hand, our experiments indeed show something interesting about making \varphi (decoder activation) different from \sigma (encoder activation)! Let us draw attention to Scheme 4 (\varphi=ReLU, \sigma=tanh) and Scheme 1 (\varphi=\sigma=ReLU), in Fig 4. Note that tanh activation is typically considered to be rather difficult. Scheme 4 is not even an EOC initialization with respect to \sigma, whereas Scheme 1 is. Despite these, we observe that:
> - Scheme 4 has lower test loss consistently.
> - Scheme 4 has smoother learning curve, suggesting that it is possible to use a higher learning rate on this scheme.
>
> These observations are intriguing. As such, we believe your question of whether having different activations is better would make a good research direction.
>
>
> ********************************************
> Question: “It would be interesting to see how this analysis is applied to Denoising Autoencoders [3], which should be straightforward to apply similar to dropout analysis appeared in Schoenholz et al [2].”
>
> Answer: We totally agree on this! We believe it would be interesting to examine the behavior of various variants of the autoencoders, which warrant multiple serious investigations.

---

### Official Review · AnonReviewer3 · 2018-11-02
**Very interesting contribution on weighted-tied random auto-encoder**

**Rating:** 9
**Confidence:** 4

**Review:**

Building on the recent progresses in the analysis of random high-dimensional statistics problem and in particular of message passing algorithm, this paper analyses the performances of weighted tied auto-encoder.  Technically, the paper is using the state evolution formalism. In particular the main theorem uses the analysis of the multi-layer version of these algorithm, the so-called state evolution technics, in order to analyse the behaviour of optimal decoding in weight-tied decoder. It is based on a clever trick that the behaviour of the decoding is similar to the one of the reconstruction on a multilayer estimation problem. This is a very orginal use of these technics.

The results are 3-folds: (i) a deep analysis of the limitation of weight-tied DAE, in the random setting, (ii) the demonstration of the sensitivity to perturbations and (iii) a clever method for initialisation that  to train a DAE.

Pro: a rigorous work, a clever use of the recent progresses in rigorous analysis of random neural net, and a very deep answer to interesting questions, and
Con: I do not see much against the paper. A minor comment: the fact that the DAE is "weight-tied" is fundamental in this analysis. It actually should be mentioned in the title!

---

> ### Author Response · Authors · 2018-11-26
> **Thank you for the encouraging review**
>
> We thank the reviewer for finding our work interesting! We completely agree that one key contribution of our work is the application of the Gaussian conditioning technique, borrowed from the TAP theory / approximate message passing literature, to the random deep weight-tied autoencoder setting.
>
> Question: “A minor comment: the fact that the DAE is "weight-tied" is fundamental in this analysis. It actually should be mentioned in the title!”
>
> Answer: Thank you for the suggestion. We agree with that and have revised the title of the paper accordingly.

---

### Meta-Review · Area_Chair1 · 2018-12-11
**ICLR 2019 decision**

**Confidence:** 4
**Recommendation:** Accept (Oral)

**Metareview:**

This paper analyzes random auto encoders in the infinite dimension limit with an assumption that the weights are tied in the encoder and decoder. In the limit the paper is able to show the random auto encoder transformation  as doing an approximate inference on data. The paper is able to obtain principled initialization strategies for training deep autoencoders using this analysis, showing the usefulness of their analysis. Even though there are limitations of paper such as studying only random models, and characterizing them only in the limit, all the reviewers agree that the analysis is novel and gives insights on an interesting problem.